# Principled Deep Neural Network Training through Linear Programming

## Abstract

Deep Learning has received significant attention due to its impressive performance in many state-of-the-art learning tasks. Unfortunately, while very powerful, Deep Learning is not well understood theoretically and in particular only recently results for the complexity of training deep neural networks have been obtained. In this work we show that large classes of deep neural networks with various architectures (e.g., DNNs, CNNs, Binary Neural Networks, and ResNets), activation functions (e.g., ReLUs and leaky ReLUs), and loss functions (e.g., Hinge loss, Euclidean loss, etc) can be trained to near optimality with desired target accuracy using linear programming in time that is exponential in the input data and parameter space dimension and polynomial in the size of the data set; improvements of the dependence in the input dimension are known to be unlikely assuming $P \neq NP$, and improving the dependence on the parameter space dimension remains open. In particular, we obtain polynomial time algorithms for training for a given fixed network architecture. Our work applies more broadly to empirical risk minimization problems which allows us to generalize various previous results and obtain new complexity results for previously unstudied architectures in the proper learning setting.

## 1 Introduction

Deep Learning is a powerful tool for modeling complex learning tasks. Its versatility allows for nuanced architectures that capture various setups of interest and has demonstrated a nearly unrivaled performance on state-of-the-art learning tasks across many domains. At the same time, the fundamental behavior of Deep Learning methods is not well understood. One particular aspect that recently gained significant interest is the computational complexity of training such networks. The basic training problem is usually formulated as an *empirical risk minimization problem (ERM)* that can be phrased as

$$\min_{\phi \in \Phi} \frac{1}{D} \sum_{i=1}^{D} \ell(f(\hat{x}^i, \phi), \hat{y}^i), \tag{1}$$

where $\ell$ is some *loss function*, $(\hat{x}^i, \hat{y}^i)_{i=1}^{D}$ is an i.i.d. sample from some data distribution $\mathcal{D}$, and $f$ is a neural network architecture parameterized by $\phi \in \Phi$ with $\Phi$ being the parameter space of the considered architecture (e.g., network weights). The empirical risk minimization problem is solved *in lieu* of the *general risk minimization problem (GRM)* $\min_{\phi \in \Phi} \mathbb{E}_{(x,y) \in \mathcal{D}} [\ell(f(x, \phi), y)]$ which is usually impossible to solve due to the inaccessibility of $\mathcal{D}$. Several works have studied the training problem for specific architectures, both in the *proper* and *improper* learning setup. In the former, the resulting "predictor" obtained from (1) is always of the form $f(\cdot, \hat{\phi})$ for some $\hat{\phi} \in \Phi$, whereas in the latter, the predictor is allowed to be outside the class of

functions $\{f(\cdot, \phi) : \phi \in \Phi\}$ as long as it satisfies certain approximation guarantee to the solution of (1)[1]. In both cases, all results basically establish trainability in time that is exponential in the network parameters but polynomial in the amount of data. In this work we complement and *significantly extend* previous work by providing a principled method to convert the empirical risk minimization problem in (1) associated with the learning problem for various architectures into a *linear programming problem (LP)* in the *proper learning* setting. The obtained linear programming formulations are of size roughly exponential in the input dimension and in the parameter space dimension and *linear* in the size of the data-set. This result provides new bounds on the computational complexity of the training problem. For an overview on Complexity Theory we refer the reader to Arora and Barak (2009).

### Related Work

Our work is most closely related to Goel et al. (2017), Zhang et al. (2016), and Arora et al. (2018). In Zhang et al. (2016) the authors show that $\ell_1$-regularized networks can be learned improperly in polynomial time in the size of the data (with a possibly exponential architecture dependent constant) for networks with ReLU-like activations (but not actual ReLUs) and an arbitrary number of layers $k$. These results were then generalized in Goel et al. (2017) to actual ReLU activations. In both cases the *improper learning* setup is considered, i.e., the learned predictor is not a neural network itself and the learning problem is solved approximately for a given target accuracy. In contrast to these works, Arora et al. (2018) considered proper and exact learning however only for $k = 2$ (i.e., one hidden layer).

In relation to these works, we consider the *proper learning* setup for an arbitrary number of layers $k$ and a wide range of activations, loss functions, and architectures. As previous works, except for Arora et al. (2018), we consider the approximate learning setup as we are solving the empirical risk minimization problem and we also establish generalization of our so-trained models. Our approach makes use of Bienstock and Muñoz (2018) that allows for reformulating non-convex optimization problems with small treewidth and discrete as well as continuous decision variables as an approximate linear programming formulations.

To the best of our knowledge, there are no previous papers that propose LP-based approaches for training neural networks. There are, however, proposed uses of Mixed-Integer and Linear Programming technology in other aspects of Deep Learning. Some examples of this include feature visualization (Fischetti and Jo, 2018), generating adversarial examples (Cheng et al., 2017; Anonymous, 2019a; Fischetti and Jo, 2018), counting linear regions of a Deep Neural Network (Serra et al., 2017), performing inference (Amos et al., 2017) and providing strong convex relaxations for trained neural networks (Anderson et al., 2018).

### Contribution

We first establish a general framework that allows us to reformulate (regularized) ERM problems arising in Deep Learning (among others!) into approximate linear programs with explicit bounds on their complexity. The resulting methodology allows for providing complexity upper bounds for specific setups simply by plugging-in complexity measures for the constituting elements such as layer architecture, activation functions, and loss functions. In particular our approach overcomes limitations of previous approaches in terms of handling the accuracy of approximations of non-linearities used in the approximation functions to achieve the overall target accuracy.

**Principled Training through LPs.** If $\epsilon > 0$ is arbitrary, then for any sample size $D$ there exists a *data-independent* linear program, i.e., the LP can be written down *before* seeing the data, with the following properties:

---

[1]In the case of Neural Networks, for example, *improper* learning could lead to a predictor that does not correspond to a Neural Network, but that might behave closely to one.

*Solving the ERM problem to $\epsilon$-optimality.* The linear program describes a polyhedron $P$ such that for every realized data set $(\hat{X}, \hat{Y}) = (\hat{x}^i, \hat{y}^i)_{i=1}^D$ there is a face $\mathcal{F}_{\hat{X}, \hat{Y}} \subseteq P$ such that optimizing certain *linear* function over $\mathcal{F}_{\hat{X}, \hat{Y}}$ solves (1) to $\epsilon$-optimality returning a feasible parametrization $\tilde{\phi} \in \Phi$ which is part of our hypothesis class (i.e., we consider *proper learning*). The face $\mathcal{F}_{\hat{X}, \hat{Y}} \subseteq P$ is simply obtained from $P$ by fixing certain variables of the linear program using the values of the actual sample; equivalently, by Farkas' lemma, this can be achieved by modifying the objective function to ensure optimization over the face belonging to the data. As such, the linear program has a *build-once-solve-many* feature. We will also show that a possible *data-dependent* LP formulation is meaningless (see Appendix B).

*Size of the linear program.* The *size*, measured as bit complexity, of the linear program is roughly $O((2\mathcal{L}/\epsilon)^{N+n+m} D)$ where $\mathcal{L}$ is a constant depending on $\ell$, $f$, and $\Phi$ that we will introduce later, $n, m$ are the dimensions of the data points, i.e., $\hat{x}^i \in \mathbb{R}^n$ and $\hat{y}^i \in \mathbb{R}^m$ for all $i \in [D]$, and $N$ is the dimension of the parameter space $\Phi$. The overall learning algorithm is obtained then by formulating and solving the linear program, e.g., with the ellipsoid method whose running time is polynomial in the size of the input Grötschel et al. (2012). Even sharper size bounds can be obtained for specific architectures assuming network structure (see Appendix F) and our approach immediately extends to regularized ERMs (see Appendix E). It is important to mention that the constant $\mathcal{L}$ measures a certain Lipschitzness of the ERM training problem. While not exactly requiring Lipschitz continuity in the same way, Lipschitz constants have been used before for measuring complexity in the *improper* learning framework (see Goel et al. (2017)) and more recently have been shown to be linked to generalization in Gouk et al. (2018).

*Generalization.* Additionally, we establish that the solutions obtained for the ERM problem via our linear programming approach generalize, utilizing techniques from stochastic optimization. We also show that using our approach one can obtain a significant improvement on the results of Goel et al. (2017) when approximating the *general risk minimization problem*. Due to space limitations, however, we relegate this discussion to Appendix I.

Throughout this work we assume both data and parameters to be well-scaled, which is a common assumption and mainly serves to simplify the representation of our results; the main assumption is the reasonable boundedness, which can be assumed without significant loss of generality as actual computations assume boundedness in any case (see also Liao et al. (2018) for arguments advocating the use of normalized coefficients in neural networks). More specifically, we assume $\Phi \subseteq [-1, 1]^N$ as well as $(x, y) \sim \mathcal{D}$ satisfies $(x, y) \in [-1, 1]^n \times [-1, 1]^m$. We point out three important features of our results. First, we provide a solution method that has provable optimality guarantees for the ERM problem, ensures generalization, and *linear* dependency on the data (in terms of the complexity of the LP) without assuming convexity of the optimization problem. To the best of our knowledge, the only result presenting optimality guarantees in a *proper learning*, non-convex setting is that of Arora et al. (2018). Second, the linear program that we construct for a given sample size $D$ is *data-independent* in the sense that it can be written down *before seeing the actual data realization* and as such it encodes reasonable approximations of all possible data sets that can be given as an input to the ERM problem. This in particular shows that our linear programs are *not* simply discretizing space: if one considers a discretization of data contained in $[-1, 1]^n \times [-1, 1]^m$, the total number of possible data sets of size $D$ is exponential in $D$, which makes the linear dependence on $D$ of the size of our LPs a remarkable feature. Finally, our approach can be directly extended to handle commonly used regularizers as we show in Appendix E; for ease of presentation though we omit regularizers throughout our main discussions.

**Complexity results for various network architectures.** We apply our methodology to various well-known neural network architectures and either generalize previous results or provide completely new results. We provide an overview of our results in Table 1, where $k$ is the number of layers, $w$ is width of the network, $n/m$ are the input/output dimensions and $N$ is the total number of parameters. We use $G$ to denote the directed graph defining the neural network and $\Delta$ the maximum vertex in-degree in $G$. In all results the node

Table 1: Summary of results for various architectures

| Type | Loss Function | Size of LP | Remarks |
|------|--------------|-----------|---------|
| Fully connected DNN | Absolute/Quadratic/Hinge | $O\left(\left(mw^{O(k^2)}/\epsilon\right)^{n+m+N} D\right)$ | $N = \|E(G)\|$ |
| Fully connected DNN | Cross-Entropy + Soft-Max | $O\left(\left(m\log(m)w^{O(k^2)}/\epsilon\right)^{n+m+N} D\right)$ | $N = \|E(G)\|$ |
| Convolutional NN | Absolute/Quadratic/Hinge | $O\left(\left(mw^{O(k^2)}/\epsilon\right)^{n+m+N} D\right)$ | $N \ll \|E(G)\|$ |
| ResNet | Absolute/Quadratic/Hinge | $O\left(\left(m\Delta^{O(k^2)}/\epsilon\right)^{n+m+N} D\right)$ | |
| ResNet | Cross-Entropy + Soft-Max | $O\left(\left(m\log(m)\Delta^{O(k^2)}/\epsilon\right)^{n+m+N} D\right)$ | |

computations are linear with bias term and normalized coefficients, and activation functions with Lipschitz constant at most 1 and with 0 as a fixed point; these include *ReLU, Leaky ReLU, eLU, Tanh*, among others.

We would like to point out that certain improvements in the results in Table 1 can be obtained by further specifying if the ERM problem corresponds to *regression* or *classification*. For example, the choice of loss functions and the nature of the output data *y* (discrete or continuous) typically rely on this. We can exploit such features in the construction of the LPs (see the proof of Theorem 3.1) and provide a sharper bound on the LP size. Nonetheless, these improvements are not especially significant and in the interest of clarity and brevity we prefer to provide a unified discussion on ERM. Missing proofs have been relegated to the appendix due to space limitations.

The complexity of the training problem for the Fully Connected DNN case is arguably the most studied and, to the best of our knowledge, all training algorithms with approximation or optimality guarantees have a polynomial dependency on *D* only *after* fixing the architecture (depth, width, input dimension, etc.). In our setting, once the architecture is fixed, we obtain a polynomial dependence in both *D* and $1/\epsilon$. Moreover, our results show that in the bounded case one can obtain a training algorithm with polynomial dependence on *D* across all architectures, assuming very little on the specific details of the network (loss, activation, etc). This answers an open question left by Arora et al. (2018) regarding the possibility of a training algorithm with polynomial dependence on *D*. In addition, we show that a *uniform* LP can be obtained without compromising that dependence on *D*.

The reader might wonder if the exponential dependence on the other parameters of our LP sizes can be improved, namely the input dimension $n + m$ and the parameter space dimension *N* (we are ignoring for the moment the exponent involving the depth *k*, as it will be typically dominated by *N*). The dependence on the input dimension is unlikely to be improved due to the NP-hardness result in Blum and Rivest (1992), whereas obtaining a polynomial dependence on the parameter space dimension remains open (see Arora et al. (2018)). A recent paper Anonymous (2019b) provides an NP-hard DNN training problem that becomes polynomially solvable when the input dimension is fixed. However, this result considers a fixed architecture, thus the parameter space dimension is a constant and the running time is measured with respect to *D*.

## 2 PRELIMINARIES

In the following let $[n] \doteq \{1, \ldots, n\}$ and $[n]_0 \doteq \{0, \ldots, n\}$. Given a graph *H*, we will use $V(H)$ and $E(H)$ to denote the vertex-set and edge-set of *H*, respectively, and $\delta_H(u)$ will be the set of edges incident to vertex *u*. We will need:

**Definition 2.1.** For a function $g : \mathcal{K} \subseteq \mathbb{R}^n \to \mathbb{R}$, we denote its *Lipschitz constant with respect to the p-norm over $\mathcal{K}$* as $\mathcal{L}_p(g)$, satisfying $|g(x) - g(y)| \leq \mathcal{L}_p(g)\|x - y\|_p$ for all $x, y \in \mathcal{K}$ (whenever it exists).

Moreover, in the following let $\mathbb{E}_{\omega \in \Omega}[\cdot]$ and $\mathbb{V}_{\omega \in \Omega}[\cdot]$ denote the *expectation* and *variance* with respect to the random variable $\omega \in \Omega$, respectively.

## 2.1 EMPIRICAL RISK MINIMIZATION

The basic ERM problem is typically of the form (1), where $\ell$ is some *loss function*, $(\hat{x}^i, \hat{y}^i)_{i=1}^D$ is an i.i.d. sample from some data distribution $\mathcal{D}$ that we have reasonable *sampling* access to, and $f$ is a model that is parametrized by $\phi \in \Phi$. We consider the proper learning setting here, where the computed solution to the ERM problem has to belong to the hypothesis class induced by $\Phi$; for a detailed discussion see Appendix A.2. We next define the Lipschitz constant of an ERM problem with respect to the infinity norm.

**Definition 2.2.** Consider the ERM problem (1) with parameters $D, \Phi, \ell, f$. We define the *Architecture Lipschitz Constant* $\mathcal{L}(D, \Phi, \ell, f)$ as

$$\mathcal{L}(D, \Phi, \ell, f) \doteq \mathcal{L}_\infty(\ell(f(\cdot, \cdot), \cdot)) \tag{2}$$

over the domain $\mathcal{K} = [-1, 1]^n \times \Phi \times [-1, 1]^m$.

We emphasize that in (2) we are considering the data-dependent entries as variables as well, and not only the parameters $\Phi$ as it is usually done in the literature. This is because we will construct data-independent LPs, a subtlety that will become clear later.

## 2.2 NEURAL NETWORKS

A neural network can be understood as a function $f$ defined over a directed graph that maps inputs $x \in \mathbb{R}^n$ to $f(x) \in \mathbb{R}^m$. The directed graph $G = (V, E)$, which represents the network architecture, often naturally decomposes into layers $V = \bigcup_{i \in [k]_0} V_i$ with $V_i \subseteq V$, where $V_0$ is referred to as the *input layer* and $V_k$ as the *output layer*. To all other layers we refer to as *hidden layers*. These graphs do neither have to be acyclic (as in the case of *recurrent neural networks*) nor does the layer decomposition imply that arcs are only allowed between adjacent layers (as in the case of *ResNets*). In *feed-forward* networks, however, the graph is assumed to be acyclic. For the unfamiliar reader we provide a more formal definition in Appendix A.3.

## 2.3 BINARY OPTIMIZATION PROBLEMS WITH SMALL TREEWIDTH

We will introduce the key concepts that we need to formulate and solve binary optimization problems with small treewidth, which will be the main workhorse behind our results. The *treewidth* of a graph is a parameter used to measure how *tree-like* a given graph is. Among all its equivalent definitions, the one we will use in this work is the following:

**Definition 2.3.** Let $G$ be an undirected graph. A *tree-decomposition* (Robertson and Seymour (1986)) of $G$ is a pair $(T, Q)$ where $T$ is a tree and $Q = \{Q_t : t \in V(T)\}$ is a family of subsets of $V(G)$ such that

(i) For all $v \in V(G)$, the set $\{t \in V(T) : v \in Q_t\}$ forms a sub-tree $T_v$ of $T$, and

(ii) For each $\{u, v\} \in E(G)$ there is a $t \in V(T)$ such that $\{u, v\} \subseteq Q_t$, i.e., $t \in T_u \cap T_v$.

The *width* of the decomposition is defined as $\max\{|Q_t| : t \in V(T)\} - 1$. The *treewidth of $G$* is the minimum width over all tree-decompositions of $G$.

We refer to the $Q_t$ as *bags* as customary. An example of a tree-decomposition is given in Figure 2 in Appendix A.1. In addition to *width*, another important feature of a tree-decomposition $(T, Q)$ we use is the *size of the tree-decomposition* given by $|V(T)|$.

Consider a problem of the form

$$(\textbf{BO}) \; : \; \min \; c^T x + d^T y$$
$$\text{s.t.} \; f_i(x) \; \geq \; 0, \, i \in [m] \quad g_j(x) \; = \; y_j, \, j \in [p]$$
$$x \in \{0, 1\}^n, y \in \mathbb{R}^p,$$

where the $f_i$ and $g_j$ are arbitrary functions that we access via a function value oracle, i.e., an oracle that returns the function upon presentation with an input. We will further use the concept of *intersection graph*.

**Definition 2.4.** The *intersection graph* $\Gamma[\mathcal{I}]$ for an instance $\mathcal{I}$ of **BO** is the undirected graph which has a vertex for each $x$ variable and an edge for each pair of $x$ variables that appear in any common constraint.

Note that in the above definition we have ignored the $y$ variables which will be of great importance later. The *sparsity of a problem* is now given by the treewidth of its intersection graph and we obtain:

**Theorem 2.5.** *Consider an instance $\mathcal{I}$ of problem **BO**. If $\Gamma[\mathcal{I}]$ has a tree-decomposition $(T, Q)$ of width $\omega$, there is an exact linear programming reformulation of $\mathcal{I}$ with $O\left(2^\omega \left(|V(T)| + p\right)\right)$ variables and constraints.*

Theorem 2.5 is an immediate generalization of a theorem in Bienstock and Muñoz (2018) distinguishing the variables $y$, which do not need to be binary in nature, but are fully determined by the binary variables $x$. The proof is omitted as it is almost identical to the proof in Bienstock and Muñoz (2018). For the sake of completeness, we include a proof sketch in Appendix C.1.

## 3 Approximation to ERM via data-independent LPs

We will now show how we can obtain an approximate LP formulation for the ERM problem. A notable feature is that our LP formulation is *data-independent* in the sense that we can write down the LP, for a given sample size $D$, *before* having seen the actual data; the LP is later specialized to a given data set by fixing some of its variables. This subtlety is extremely important as it prevents trivial solutions, where some non-deterministic guess provides a solution to the ERM problem for a *given* data set and then simply writes down a small LP that outputs the network configuration; such an LP would be of small size (the typical notion of complexity used for LPs) however not efficiently computable. By making the construction independent of the data we circumvent this issue; we provide a short discussion in Appendix B and refer the interested reader to Braun et al. (2016; 2015); Braun and Pokutta (2018+) for an in-depth discussion of this subtlety. Slightly simplifying, we might say for now that in the same way we do not want algorithms to be designed for a *fixed data set*, we do not want to construct LPs for a specific data set but for a wide range of data sets.

As mentioned before, we assume $\Phi \subseteq [-1, 1]^N$ as well as $(x, y) \sim \mathcal{D}$ satisfies $(x, y) \in [-1, 1]^n \times [-1, 1]^m$ as normalization to simplify the exposition. Since the **BO** problem only considers linear objective functions, we begin by reformulating the ERM problem (1) in the following equivalent form:

$$\min_{\phi \in \Phi} \left\{ \frac{1}{D} \sum_{d=1}^{D} L_d \; \middle| \; L_d \; = \; \ell(f(\hat{x}^d, \phi), \hat{y}^d) \quad \forall \, d \in [D] \right\} \tag{4}$$

### 3.1 Approximation of the feasible region via an $\epsilon$-grid

Motivated by this reformulation, we study an approximation to the following set:

$$S(D, \Phi, \ell, f) = \left\{ (x^1, \ldots, x^D, y^1, \ldots, y^D, \phi, L) \in [-1, 1]^{(n+m)D} \times \Phi \times \mathbb{R}^D \; : \; L_d = \ell(f(x^d, \phi), y^d) \right\} \tag{5}$$

The variables $(x^i, y^i)_{i=1}^D$ denote the data variables, that will be assigned values upon a specification of a data set of sample size $D$.

Let $r \in \mathbb{R}$ with $-1 \le r \le 1$. Given $\gamma \in (0, 1)$ we can approximate $r$ as a sum of inverse powers of 2, within additive error proportional to $\gamma$. For $N_\gamma \doteq \lceil \log_2 \gamma^{-1} \rceil$ there exist values $z_h \in \{0, 1\}$ with $h \in [N_\gamma]$, so that

$$-1 + 2 \cdot \sum_{h=1}^{N_\gamma} 2^{-h} z_h \quad \le \quad r \quad \le \quad -1 + 2 \cdot \sum_{h=1}^{N_\gamma} 2^{-h} z_h + 2\gamma \le 1. \tag{6}$$

Our strategy is now to approximately represent the $x, y, \phi$ variables via these binary approximations, i.e., as $-1 + 2 \cdot \sum_{h=1}^{L_\gamma} 2^{-h} z_h$ where each $z_h$ is a (new) binary variable. Define $\epsilon = 2\gamma\mathcal{L}$, where $\mathcal{L} = \mathcal{L}(D, \Phi, \ell, f)$ is the architecture Lipschitz constant defined in (2), and consider the following approximation of $S(D, \Phi, \ell, f)$:

$$S^\epsilon(D, \Phi, \ell, f) \doteq \Big\{(x^1, \ldots, x^D, y^1, \ldots, y^D, \phi, L) \in [-1, 1]^{(n+m)D} \times \Phi \times \mathbb{R}^D \; : \; z \in \{0, 1\}^{N_\gamma(N+Dn+Dm)}$$

$$L_d = \ell(f(x^d, \phi), y^d), \, d \in [D] \qquad\qquad \phi_i = -1 + 2\sum_{h=1}^{N_\gamma} 2^{-h} z_{i,h}^\phi, \, i \in [N]$$

$$y_i^d = -1 + 2\sum_{h=1}^{N_\gamma} 2^{-h} z_{i,h}^{y^d}, \, d \in [D], \, i \in [m] \qquad x_i^d = -1 + 2\sum_{h=1}^{N_\gamma} 2^{-h} z_{i,h}^{x^d}, \, d \in [D], \, i \in [n]\Big\}.$$

We can readily describe the error of the approximation of $S(D, \Phi, \ell, f)$ by $S^\epsilon(D, \Phi, \ell, f)$ in the ERM problem (1) induced by the discretization:

**Lemma 3.1.** *For any $(x^1, \ldots, x^D, y^1, \ldots, y^D, \phi, L) \in S(D, \Phi, \ell, f)$ there is $(\hat{x}^1, \ldots, \hat{x}^D, \hat{y}^1, \ldots, \hat{y}^D, \hat{\phi}, \hat{L}) \in S^\epsilon(D, \Phi, \ell, f)$ such that $\left|\frac{1}{D}\sum_{d=1}^D L_d - \frac{1}{D}\sum_{d=1}^D \hat{L}_d\right| \le \epsilon$.*

By substituting out the $x, y, \phi$ by means of the equations of $S^\epsilon(D, \Phi, \ell, f)$, we obtain a feasible region as **BO**.

## 3.2 Linear reformulation of the binary approximation

So far, we have phrased the ERM problem (1) in terms of a binary optimization problem using a discretization of the continuous variables. This in and of itself is neither insightful nor useful. In this section we will perform the key step, reformulating the convex hull of $S^\epsilon(D, \Phi, \ell, f)$ as a moderate-sized linear program by means of Theorem 2.5 exploiting small treewidth of the ERM problem.

After replacing the $(x, y, \phi)$ variables in $S^\epsilon(D, \Phi, \ell, f)$ using the $z$ variables, we can see that the intersection graph of $S^\epsilon(D, \Phi, \ell, f)$ is given by Figure 1a, where we use $(x, y, \phi)$ as stand-ins for corresponding the binary variables $z^x, z^y, z^\phi$. Recall that the intersection graph does not include the $L$ variables. It is not hard to see that a valid tree-decomposition for this graph is given by Figure 1b. This tree-decomposition has size $D$ and width $N_\gamma(n + m + N) - 1$ (much less than the $N_\gamma(N + Dn + Dm)$ binary variables). This yields our main theorem:

**Main Theorem 3.1.** *Let $D \in \mathbb{N}$ be a given sample size. Then $\mathrm{conv}(S^\epsilon(D, \Phi, \ell, f))$ admits a linear programming formulation with the following properties:*

(a) *The linear program has no more than $4D(2\mathcal{L}/\epsilon)^{n+m+N}$ variables and $2D(2(2\mathcal{L}/\epsilon)^{n+m+N}+1)$ constraints. We refer to the resulting polytope as $P_{S_\epsilon}$.*

(b) *The linear program can be constructed in time $O((2\mathcal{L}/\epsilon)^{n+m+N} D)$ plus the time required for $O((2\mathcal{L}/\epsilon)^{n+m+N})$ evaluations of $\ell$ and $f$.*

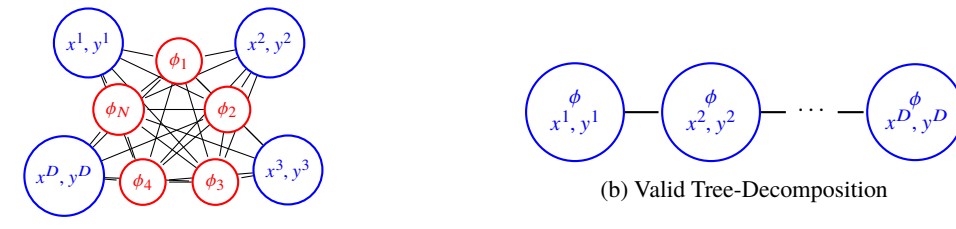

(a) Intersection Graph of $S^\epsilon(D, \Phi, \ell, f)$

(b) Valid Tree-Decomposition

Figure 1: Intersection Graph and Tree-Decomposition of $S^\epsilon(D, \Phi, \ell, f)$

(c) *Fix any data set* $(\hat{X}, \hat{Y}) = (\hat{x}^i, \hat{y}^i)_{i=1}^D$ *of size D, with* $\hat{x}^i \in [-1, 1]^n$ *and* $\hat{y}^i \in [-1, 1]^m$. *There exists a face* $\mathcal{F}_{\hat{X}, \hat{Y}}$ *of* $P_{S_\epsilon}$ *such that*

$$\tilde{\phi} \in \operatorname*{argmin}_{\phi, L} \left\{ \frac{1}{D} \sum_{i=1}^D L_i \ \middle| \ (\phi, L) \in \operatorname*{proj}_{\phi, L}(\mathcal{F}_{\hat{X}, \hat{Y}}) \right\}$$

*satisfies* $\left| \frac{1}{D} \sum_{i=1}^D \left[ \ell(f(\hat{x}^i, \phi^*), \hat{y}^i) - \ell(f(\hat{x}^i, \tilde{\phi}), \hat{y}^i) \right] \right| \le 2\epsilon$, *where* $\phi^* \in [-1, 1]^N$ *is an optimal solution to the ERM problem* (1) *with input data* $(\hat{X}, \hat{Y})$. *This means that solving an LP using an appropriate face of* $P_{S_\epsilon}$ *solves the ERM problem* (1) *within an additive error* $2\epsilon$.

(d) *The face* $\mathcal{F}_{\hat{X}, \hat{Y}}$ *arises by substituting in actual data for the data-variables* $x, y$, *which determine the approximations* $z^x, z^y$ *and is used to fixed additional variables of the LP.*

*Proof.* The proof of part (a) follows directly from Theorem 2.5 using $N_\gamma = \lceil \log(2\mathcal{L}/\epsilon) \rceil$ along with the tree-decomposition of Figure 1b, which implies $|V(T')| + p = 2D$ in this case. Parts (b), (c) and (d) rely on the explicit construction for Theorem 2.5 and they are given in Appendix D. □

Observe that equivalently, by Farkas' lemma, optimizing over the face can be also achieved by modifying the objective function in a straightforward way. Also note that the number of evaluations of $\ell$ and $f$ is independent of $D$. We would like to further point out that we can provide an interesting refinement of the theorem from above: if $\Phi$ has an inherent network structure (as in the case of Neural Networks) one can exploit treewidth-based sparsity of the *network itself* in order to obtain a smaller linear program with the same approximation guarantees as before. This allows us to reduce the exponent in the exponential term of the LP size to an expression that depends on the *sparsity of the network*, instead of its size. For brevity of exposition, we relegate this discussion to Appendix F. Another improvement can be obtained by using more information about the input data. Assuming extra structure on the input, one could potentially improve the $n + m$ exponent on the LP size. We relegate the discussion of this feature to Remark D.1 in the Appendix, as it requires the explicit construction of the LP, which we also provide in the Appendix.

## 4 COMPLEXITY FOR SPECIFIC ARCHITECTURES

### 4.1 FULLY-CONNECTED LAYERS WITH ReLU ACTIVATIONS AND NORMALIZED COEFFICIENTS

We consider a Deep Neural Network $f : \mathbb{R}^n \to \mathbb{R}^m$ with $k$ layers given by $f = T_k \circ \sigma \circ \cdots \circ T_2 \circ \sigma \circ T_1$, where $\sigma$ is the ReLU activation function $\sigma(x) \doteq \max\{0, x\}$ applied component-wise and each $T_i : \mathbb{R}^{w_{i-1}} \to \mathbb{R}^{w_i}$ is an affine linear function. Here $w_0 = n$ is the dimension of the input data and $w_k = m$ is the dimension of the output of the network. We write $T_i(z) = A_i z + b_i$ for $i \in [k]$ and assume $\|A_i\|_\infty \le 1$, $\|b_i\|_\infty \le 1$ via

normalization. Thus, if $v$ is a node in layer $i$, the node computation performed in $v$ is of the form $\hat{a}^T z + \hat{b}$, where $\hat{a}$ is a row of $A_i$ and $\hat{b}$ is a component of $b_i$. Note that in this case the dimension of the parameter space $\Phi$ is exactly the number of edges of the network. Hence, we use $N$ to represent the number of edges as well. We begin with a short technical Lemma, with which we can immediately establish the following corollary.

**Lemma 4.1.** *For every $i \in [k-1]_0$ define $U_i = \sum_{j=0}^{i} w^j$. If $\|z\|_\infty \leq U_i$ then $\|T_{i+1}(z)\|_\infty \leq U_{i+1}$.*

**Corollary 4.2.** *If $\Phi$ is the class of Neural Networks with $k$ layers, $N$ edges, ReLU activations, and normalized coefficients, then $\mathrm{conv}(S^\epsilon(D, \Phi, \ell, f))$ can be formulated via a linear program of size $O((2\mathcal{L}_\infty(\ell)w^{O(k^2)}/\epsilon)^{n+m+N} D)$, where $w = \max_{i \in [k-1]_0} w_i$ and $\mathcal{L}_\infty(\ell)$ is the Lipschitz constant of $\ell(\cdot, \cdot)$ over $[-U_k, U_k]^m \times [-1, 1]^m$. The linear program can be constructed in time $O((2\mathcal{L}_\infty(\ell)w^{O(k^2)}/\epsilon)^{n+m+N} D)$ plus the time required for $O((2\mathcal{L}_\infty(\ell)w^{O(k^2)}/\epsilon)^{n+m+N})$ evaluations of $\ell$ and $f$.*

*Proof.* Proving that the architecture Lipschitz constant is $L_\infty(\ell)w^{O(k^2)}$ suffices. Note that all node computations take the form $h(z, a, b) = z^T a + b$ for $a \in [-1, 1]^w$ and $b \in [-1, 1]$. The only difference is made in the domain of $z$, which varies from layer to layer. The 1-norm of the gradient of $h$ is at most $\|z\|_1 + \|a\|_1 + 1 \leq \|z\|_1 + w + 1$ which, in virtue of Lemma 4.1, implies that a node computation on layer $i$ (where the weights are considered variables as well) has Lipschitz constant at most $\sum_{j=0}^{i} w^j + w + 1 \leq \sum_{j=0}^{i} 2w^j = 2\frac{w^{i+1}-1}{w-1} \leq 2w^{i+1}$. On the other hand, for $\|(a, b) - (a', b')\|_\infty \leq \gamma$ and $z \in [-U_i, U_i]$, it holds that $|h(z, a, b) - h(z', a', b')| \leq 2w^{i+1}\|(z - z', a - a', b - b')\|_\infty \leq 2w^{i+1} \max\{\|z - z'\|_\infty, \gamma\}$, which shows that the Lipschitz constants can be multiplied layer-by-layer to obtain the overall architecture Lipschitz constant. Since ReLUs have Lipschitz constant equal to 1, and $\prod_{i=1}^{k} 2w^{i+1} = w^{O(k^2)}$, whenever $w \geq 2$, we conclude the architecture Lipschitz constant is $L_\infty(\ell)w^{O(k^2)}$. $\square$

The reader might have noticed that a sharper bound for the Lipschitz constant above could have been used, however we chose simpler bounds for the sake of presentation. It is worthwhile to compare the previous lemma to the following closely related result.

**Theorem 4.3.** *(Arora et al., 2018, Theorem 4.1) Let $\Phi$ be the class of Neural Networks with 1 hidden layer ($k = 2$), convex loss function $\ell$, ReLU activations and output dimension $m = 1$. There exists an algorithm to find a global optimum of the ERM problem in time $O(2^w D^{nw} \mathrm{poly}(D, n, w))$.*

*Remark* 4.4. We point out a few key differences of this result with the algorithm we can obtain from solving the LP in Corollary 4.2: (a) One advantage of our result is the benign dependency on $D$. An algorithm that solves the training problem using our proposed LP has polynomial dependency on the data-size regardless of the architecture. (b) As we have mentioned before, our approach is able to construct an LP before seeing the data. (c) The dependency on $w$ of our algorithm is also polynomial. To be fair, we are including an extra parameter $N$—the number of edges of the Neural Network—on which the size of our LP depends exponentially. (d) We are able to handle any output dimension $m$ and any number of layers $k$. (e) We do not assume convexity of the loss function $\ell$, which causes the resulting LP size to depend on how well behaved $\ell$ is in terms of its Lipschitzness. (f) The result of Arora et al. (2018) has two advantages over our result: there is no boundedness assumption on the coefficients, and they are able to provide a *globally optimal* solution instead of an $\epsilon$-approximation.

## 4.2 RESNETS, CNNS, AND ALTERNATIVE ACTIVATIONS

Corollary 4.2 can be directly generalized to handle other architectures as well, as the key features we used before are the acyclic structure of a standard Neural Network and the well behaved Lipschitz constant of the ReLU function. These features are present in many other architectures, and yield the following result.

**Lemma 4.5.** *Let $\Phi$ denote the class of feed-forward Neural Networks with $k$ layers, as defined in Section 2.2, $N$ edges, affine node computations, activation functions $a_i : \mathbb{R} \to \mathbb{R}$ with Lipschitz constant at most 1 and such that $a(0) = 0$, and normalized coefficients. Then $\text{conv}(S^\epsilon(D, \Phi, \ell, f))$ can be formulated via a linear program of size $O((2\mathcal{L}_\infty(\ell)\Delta^{O(k^2)}/\epsilon)^{n+m+N}D)$, where $\Delta$ denotes the maximum vertex in-degree of the network and $\mathcal{L}_\infty(\ell)$ is the Lipschitz constant of $\ell(\cdot, \cdot)$ over $[-U_k, U_k]^m \times [-1, 1]^m$. The linear program can be constructed in time $O((2\mathcal{L}_\infty(\ell)\Delta^{O(k^2)}/\epsilon)^{n+m+N}D)$ plus the time required for $O((2\mathcal{L}_\infty(\ell)\Delta^{O(k^2)}/\epsilon)^{n+m+N})$ evaluations of $\ell$ and $f$.*

**Corollary 4.6.** *The ERM problem (1) over Deep Residual Networks (ResNets) with 1-Lipschitz activations can be solved to $\epsilon$-optimality in time $\text{poly}(\Delta, 1/\epsilon, D)$ whenever the network size and number of layers are fixed.*

Another interesting point can be made with respect to Convolutional Neural Networks (CNN). In these, convolutional layers are included which help to significantly reduce the number of parameters involved in the neural network. From a theoretical perspective, a CNN can be obtained by simply enforcing certain parameters of a fully-connected DNN to be equal. This implies that Lemma 4.5 can also be applied to CNNs, with the key difference residing in parameter $N$, which is the dimension of the parameter space and *does not* correspond to the number of edges in a CNN. In Table 1 we provide explicit LP sizes for common architectures. These results can be directly obtained from Lemma 4.5, using the specific Lipschitz constants of the loss functions. We provide explicit computations in Appendix G.

## 5    Conclusion and Final remarks

We have presented a novel framework which shows that training a wide variety of neural networks can be done in time which depends polynomially on the data set size, while satisfying a predetermined arbitrary optimality tolerance. Our approach is realized by approaching training through the lens of linear programming. Moreover, we show that training using a particular data set is closely related to the face structure of a *data-independent* polytope.

Our contributions not only improve the best known algorithmic results for neural network training with optimality/approximation guarantees, but also shed new light on (theoretical) neural network training by bringing together concepts of graph theory, polyhedral geometry, and non-convex optimization as a tool for Deep Learning.

While the LPs we are constructing are large, and likely to be difficult to solve by straightforward use of our formulation, we strongly believe the theoretical foundations we lay here can also have practical implications in the Machine Learning community. First of all, we emphasize that all our architecture dependent terms are worst-case bounds, which can be improved by assuming more structure in the corresponding problems. Additionally, the history of Linear Programming has provided many interesting cases of extremely large LPs that can be solved to near-optimality without necessarily generating the complete LP description. In these cases the theoretical understanding of the LP structure is crucial to drive the development of incremental solution strategies.

Finally, we would like to point out an interesting connection between the way our approach works and the current most practically effective training algorithm: stochastic gradient descent. Our LP approach implicitly "decomposes" the problem for each data point, and the LP merges them back together without losing any information nor optimality guarantee, even in the non-convex setting. This is the core reason why our LP has a linear dependence on $D$, and bears close resemblance to SGD where single data points (or batches of those) are used in a given step. As such, our results might provide a new perspective, through low treewidth, on why the current practical algorithms work so well, and perhaps hints at a synergy between the two approaches. We believe this can be an interesting path to bring our ideas to practice.

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

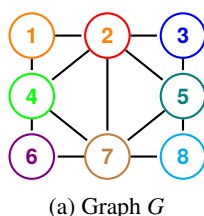

(a) Graph $G$

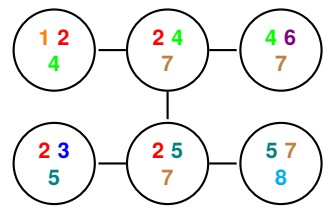

(b) A tree-decomposition of $G$ of width 2, with the sets $Q_t$ indicated inside each node of the tree.

Figure 2: Example of graph and valid tree-decomposition

## A  FURTHER DEFINITIONS

### A.1  COMPLEMENTARY DISCUSSION ON TREEWIDTH

The concept of *treewidth* is an important concept in the context of solving optimization problems with 'sparse' structure. An alternative definition to Definition 2.3 of treewidth that the reader might find useful is the following; recall that a *chordal* graph is a graph where every induced cycle has length exactly 3.

**Definition A.1.** An undirected graph $G = (V, E)$ has *treewidth* $\leq \omega$ if there exists a chordal graph $H = (V, E')$ with $E \subseteq E'$ and clique number $\leq \omega + 1$.

$H$ in the definition above is sometimes referred to as a *chordal completion* of $G$. In Figure 2 we present an example of a graph and a valid tree-decomposition. The reader can easily verify that the conditions of Definition 2.3 are met in this example. Moreover, using Definition A.1 one can verify that the treewidth of the graph in Figure 2 is exactly 2.

Two important folklore results we use in Section C.1 and Section F are the following.

**Lemma A.2.** *Let $G$ be a graph with a valid tree-decomposition $(T, Q)$ of width $\omega$. Then there exists a valid tree-decomposition $(T', Q')$ of width at most $\omega$ such that $|V(T')| \in O(|V(G)|)$.*

**Lemma A.3.** *Let $G$ be a graph with a valid tree-decomposition $(T, Q)$ and $K \subseteq V(G)$ a clique of $G$. Then there exists $t \in T$ such that $K \subseteq Q_t$.*

### A.2  PROPER VS. IMPROPER LEARNING

An important distinction is the *type* of solution to the ERM that we allow. In *proper* learning we require the solution to satisfy $\phi \in \Phi$, i.e., the model has to be from the considered model class induced by $\Phi$ and takes the form $f(\cdot, \phi^*)$ for some $\phi^* \in \Omega$, with

$$\frac{1}{D} \sum_{i=1}^{D} \ell(f(\hat{x}^i, \phi^*), \hat{y}^i) \leq \min_{\phi \in \Phi} \frac{1}{D} \sum_{i=1}^{D} \ell(f(\hat{x}^i, \phi), \hat{y}^i),$$

and this can be relaxed to $\epsilon$-approximate (proper) learning by allowing for an additive error $\epsilon > 0$ in the above. In contrast, in *improper* learning we allow for a model $g(\cdot)$, that cannot be obtained as $f(\cdot, \phi)$ with $\phi \in \Phi$, satisfying

$$\frac{1}{D} \sum_{i=1}^{D} \ell(g(\hat{x}^i), \hat{y}^i) \leq \min_{\phi \in \Phi} \frac{1}{D} \sum_{i=1}^{D} \ell(f(\hat{x}^i, \phi), \hat{y}^i),$$

with a similar approximate version. As we mentioned in the main body, this article considers the proper learning setup.

### A.3 Neural Networks

In a Neural Network, the graph $G$ defining the network can be partitioned in layers. This means that $V(G) = \bigcup_{i=0}^{k} V_i$ for some sets $V_i$ —the layers of the network. Each vertex $v \in V_i$ with $i \in [k]_0$ has an associated set of *in-nodes* denoted by $\delta^+(v) \subseteq V$, so that $(w, v) \in E$ for all $w \in \delta^+(v)$ and an associated set of *out-nodes* $\delta^-(v) \subseteq V$ defined analogously. If $i = 0$, then $\delta^+(v)$ are the *inputs* (from data) and if $i = k$, then $\delta^-(v)$ are the *outputs* of the network. Moreover, each node $v \in V$ performs a *node computation* $g_i(\delta^+(v))$, where $g_i : \mathbb{R}^{|\delta^+(v)|} \rightarrow \mathbb{R}$ with $i \in [k]$ is typically a smooth function (often these are linear or affine linear functions) and then the *node activation* is computed as $a_i(g_i(\delta^+(v)))$, where $a_i : \mathbb{R} \rightarrow \mathbb{R}$ with $i \in [k]$ is a (not necessarily smooth) function (e.g., ReLU activations of the form $a_i(x) = \max\{0, x\}$) and the value on all out-nodes $w \in \delta^-(v)$ is set to $a_i(g_i(\delta^+(v)))$ for nodes in layer $i \in [k]$. In *feed-forward* networks, we can further assume that if $v \in V_i$, then $\delta^+(v) \subseteq \bigcup_{j=0}^{i-1} V_j$, i.e., all arcs move *forward* in the layers.

## B Further discussion

### Data-dependent vs. independent LPs

As mentioned before, the assumption that the construction of the LP is independent of the specific data is important and reasonable as it basically prevents us from constructing an LP *for a specific data set*, which would be akin to designing an algorithm for a specific data set in ERM problem (1). To further illustrate the point, suppose we would do the latter, then a correct algorithm would be a simple `print` statement of the optimal configuration $\bar{\phi}$. Clearly this is nonsensical and we want the algorithm to work for all types of data sets as inputs. We have a similar requirement for the construction of the LP, with the only difference that number of data points $D$ has to be known at time of construction. As such LPs more closely resemble a circuit model of computation (similar to the complexity class P/poly); see Braun et al. (2016; 2015); Braun and Pokutta (2018+) for details.

The curious reader might still wonder how our main result changes if we allow the LPs in Theorem 3.1 to be *data-dependent*, i.e., if we construct a specific linear program *after* we have seen the data set:

*Remark* B.1. To obtain a data-dependent linear program we can follow the same approach as in Section 3 and certainly produce an LP that will provide the same approximation guarantees for a fixed data set. Moreover, since the construction of Theorem 2.5 explained in Appendix C.1 involves an enumeration over a discretization of $\Phi \subseteq [-1, 1]^N$, one can compute the (approximately) optimal solution in advance and solve this *data-dependent* ERM problem as a trivial LP with $N$ constraints, each constraint simply fixing a variable to a value; the analog to the `print` statement from above. The time needed to generate such an LP is $O((2\mathcal{L}/\epsilon)^N)$ (the number of possible discretized configurations) via at most $O((2\mathcal{L}/\epsilon)^N D)$ evaluations of $\ell$ and $f$ (one per each enumerated configuration and data-point).

This result is not particularly insightful, as it is based on a straight-forward enumeration which takes a significant amount of time, considering that it only serves one data set. On the other hand, our result shows that by including the input data as a *variable*, we do not induce an exponential term in the size of the data set $D$ and we can keep the number function evaluations to be roughly the same.

### Similarities with SGD-based training

Our approach shares some similarities with stochastic gradient descent (SGD) based training: data points are considered separately (or in small batches) and the method (in case of SGD) or the LP (in our case) ensure that the information gained from a single data point is integrated into the overall ERM solution. In the case of SGD this happens through sequential updates of the form $x_{t+1} \leftarrow x_t - \eta \nabla f_i(x_t)$, where $i$ is a random function corresponding to a training data point $(\hat{X}_i, \hat{Y}_i)$ from the ERM problem. In our case, it is the LP that 'glues

together' solutions obtained from single training data points by means of leveraging the low treewidth. This is reflected in the linear dependence in $D$ in the problem formulation size.

## C    MISSING PROOFS

*Proof of Lemma 3.1.* Choose binary values $\tilde{z}$ so as to attain the approximation for variables $x, y, \phi$ as in (6) and define $\hat{x}, \hat{y}, \hat{\phi}, \hat{L}$ from $\tilde{z}$ according to the definition of $S^{\epsilon}(D, \Phi, \ell, f)$. Since

$$\left\|(x^d, y^d, \phi) - (\hat{x}^d, \hat{y}^d, \hat{\phi})\right\|_{\infty} \leq 2\gamma = \frac{\epsilon}{\mathcal{L}} \quad d \in [D]$$

by Lipschitzness we obtain $|L_d - \hat{L}_d| \leq \epsilon$. The result then follows.    □

*Proof of Lemma 4.1.* The result can be verified directly, since for $a \in [-1, 1]^w$ and $b \in [-1, 1]$ it holds $|z^T a + b| \leq w\|z\|_{\infty} + 1$.    □

*Proof of Lemma 4.5.* The proof follows almost directly from the proof of Corollary 4.2. The two main differences are (1) the input dimension of a node computation, which can be at most $\Delta$ instead of $w$ and (2) the fact that an activation function $a$ with Lipchitz constant 1 and that $a(0) = 0$ satisfies $|a(z)| \leq |z|$, thus the domain of each node computation computed in Lemma 4.1 applies. The layer-by-layer argument can be applied as the network is feed-forward.    □

### C.1    PROOF OF THEOREM 2.5

Let us recall the definition of BO:

$$
\begin{aligned}
(\textbf{BO}) \; : \; \min \; & c^T x + d^T y \\
\text{subject to :} \quad & f_i(x) \; \geq \; 0 & & i \in [m] \\
& g_j(x) \; = \; y_j & & j \in [p] \\
& x \in \{0, 1\}^n,
\end{aligned}
$$

We sketch the proof of

**Theorem 2.5.** *Consider an instance $\mathcal{I}$ of problem **BO**. If $\Gamma[\mathcal{I}]$ has a tree-decomposition $(T, Q)$ of width $\omega$, there is an exact linear programming reformulation of $\mathcal{I}$ with $O(2^{\omega}(|V(T)| + p))$ variables and constraints.*

*Proof.* Since the support of each $f_i$ induces a clique in the intersection graph, there must exist a bag $Q$ such that supp$(f_i) \subseteq Q$ (Lemma A.3). The same holds for each $g_j$. We modify the tree-decomposition to include the $y_j$ variables the following way:

- For each $j \in [p]$, choose a bag $Q$ containing supp$(g_j)$ and add a new bag $Q'(j)$ consisting of $Q \cup \{y_j\}$ and connected to $Q$.

- We do this for every $j \in [p]$, with a different $Q'(j)$ for each different $j$. This creates a new tree-decomposition $(T', Q')$ of width at most $\omega + 1$, which has each variable $y_j$ contained in a *single* bag $Q'(j)$ which is a *leaf*.

- The size of the tree-decomposition is $|T'| = |T| + p$.

From here, we proceed as follows:

- For each $t \in T'$, if $Q'_t \ni y_j$ for some $j \in [p]$, then we construct

$$\mathcal{F}_t \doteq \{(x, y) \in \{0, 1\}^{Q_t} \times \mathbb{R} \ : \ y = g_j(x), f_i(x) \geq 0 \text{ for supp}(f_i) \subseteq Q'_t\}$$

otherwise we simply construct

$$\mathcal{F}_t \doteq \{x \in \{0, 1\}^{Q_t} \ : \ f_i(x) \geq 0 \text{ for supp}(f_i) \subseteq Q'_t\}.$$

Note that these sets have size at most $2^{|Q'_t|}$.

- We define variables $X[Y, N]$ where $Y, N$ form a partition of $Q'_{t_1} \cap Q'_{t_2}$. These are at most $2^\omega |V(T')|$.

- For each $t \in T'$ and $v \in \mathcal{F}_t$, we create a variable $\lambda_v$. These are at most $2^\omega |V(T')|$.

We formulate the following *linear* optimization problem

$$
\begin{aligned}
(\textbf{LBO}) \ : \ &\min \ c^T x + d^T y \\
\text{subject to} : \quad & \sum_{v \in \mathcal{F}_t} \lambda_v = 1 && \forall t \in T' \\
& X[Y, N] = \sum_{v \in \mathcal{F}_t} \lambda_v \prod_{i \in Y} v_i \prod_{i \in N}(1 - v_i) && \forall (Y, N) \subseteq Q'_t, \ t \in T' \\
& \lambda_v \geq 0 && \forall v \in \mathcal{F}_t, \ t \in T' \\
& x_i = \sum_{v \in \mathcal{F}_t} \lambda_v v_i && \forall i \in Q'_t \cap [n], \ t \in T' \\
& y_j = \sum_{v \in \mathcal{F}_{Q'(j)}} \lambda_v g_j(v) && \forall j \in [p]
\end{aligned}
$$

Note that the notation in the last constraint is justified since by construction $\text{supp}(g_j) \subseteq Q'(j)$. The proof of the fact that **LBO** is equivalent to **BO** follows from the arguments in Bienstock and Muñoz (2018). The key difference justifying the addition of the $y$ variables relies in the fact that they only appear in leaves of the tree decomposition $(T', Q')$, and thus in no intersection of two bags. The gluing argument using variables $X[Y, N]$ then follows directly, as it is then only needed for the $x$ variables to be binary.

We can substitute out the $x$ and $y$ variables and obtain an LP whose variables are only $\lambda_v$ and $X[Y, N]$. This produces an LP with at most $2 \cdot 2^\omega |V(T')|$ variables and $(2 \cdot 2^\omega + 1)|V(T')|$ constraints. This proves the size of the LP is $O(2^\omega(|V(T)| + p))$ as required. $\qquad \square$

# D  Data-dependent Faces and Construction Time

## D.1  Construction Time in Theorem 3.1

In this Section we show how to construct the polytope in Theorem 3.1. We first recall the following definition:

$$S(D, \Phi, \ell, f) = \{(x^1, \ldots, x^D, y^1, \ldots, y^D, \phi, L) \in [-1, 1]^{(n+m) \times D} \times \Phi \times \mathbb{R}^D \ : \ L_d = \ell(f(x^d, \phi), y^d)\}$$

and recall that $S^\epsilon(D, \Phi, \ell, f)$ is a discretized version of the set mentioned above. From the tree-decomposition detailed in Section 3.2, we see that data-dependent variables $x, y, L$ are partitioned in different bags for each

data $d \in [D]$. Let us index the bags using $d$. Since all data variables have the same domain, the sets $\mathcal{F}_d$ we construct in the proof of Theorem 2.5 will be the same for all $d \in [D]$. Using this observation, we can construct the LP as follows:

1. Fix, say, $d = 1$ and enumerate all binary vectors corresponding to the discretization of $x^1, y^1, \phi$.
2. Compute $\ell(f(x^1, \phi), y^1)$. This will take $O((2\mathcal{L}/\epsilon)^{n+m+N})$ function evaluations of $f$ and $\ell$. This defines the set $\mathcal{F}_1$.
3. Duplicate this set $D$ times, and associate each copy with a bag indexed by $d \in [D]$.
4. For each $d \in [D]$, and each $v \in \mathcal{F}_d$ create a variable $\lambda_v$.
5. For each $d \in [D-1]$, create variables $X[Y, N]$ corresponding to the intersection of bags $d$ and $d + 1$. This will create $O((2\mathcal{L}/\epsilon)^N)$ variables, since the only variables in the intersections are the discretized $\phi$ variables.
6. Formulate **LBO**.

The only evaluations of $\ell$ and $f$ are performed in the construction of $\mathcal{F}_1$. As for the additional computations, the bottleneck lies in creating all $\lambda$ variables, which takes time $O((2\mathcal{L}/\epsilon)^{n+m+N}D)$.

*Remark* D.1. Note that in step 1 of the LP construction we are enumerating all possible discretized values of $x^1, y^1$, i.e., we are implicitly assuming all points in $[-1, 1]^{n+m}$ are possible inputs. This is reflected in the $(2\mathcal{L}/\epsilon)^{n+m}$ term in the LP size estimation. If one were to use another discretization method (or a different "point generation" technique) using more information about the input data, this term could be improved and the explicit exponential dependency on the input dimension of the LP size could be alleviated significantly. However, note that in a fully-connected neural network we have $N \geq n + m$ and thus an implicit exponential dependency on the input dimension could remain unless more structure is assumed. This is in line with the NP-hardness results. We leave the full development of this potential improvement for future work.

### D.2 DATA-DEPENDENT FACES OF THE DATA-INDEPENDENT POLYTOPE

Consider a fixed data set $(\hat{X}, \hat{Y}) = (\hat{x}^i, \hat{y}^i)_{i=1}^D$ and let $\phi^*$ be an optimal solution to the ERM problem with input data $(\hat{X}, \hat{Y})$. Consider now binary variables $z^{\hat{x}}, z^{\hat{y}}$ to attain the approximation (6) and define $\tilde{x}, \tilde{y}$ from $z^{\hat{x}}, z^{\hat{y}}$, i.e., $\tilde{x}_i^d = -1 + 2\sum_{h=1}^{N_\gamma} 2^{-h} z_{i,h}^{\hat{x}d}$ and similarly for $\tilde{y}$. Define the set

$$S(\tilde{X}, \tilde{Y}, \Phi, \ell, f) = \{(\phi, L) \in \Phi \times \mathbb{R}^D : L_d = \ell(f(\tilde{x}^d, \phi), \tilde{y}^d)\}$$

and similarly as before define $S^\epsilon(\tilde{X}, \tilde{Y}, \Phi, \ell, f)$ to be the discretized version (on variables $\phi$). The following Lemma shows the quality of approximation to the ERM problem obtained using $S(\tilde{X}, \tilde{Y}, \Phi, \ell, f)$ and subsequently $S^\epsilon(\tilde{X}, \tilde{Y}, \Phi, \ell, f)$.

**Lemma D.2.** *For any $(\phi, L) \in S(\tilde{X}, \tilde{Y}, \Phi, \ell, f)$ there exists $(\phi', L') \in S^\epsilon(\tilde{X}, \tilde{Y}, \Phi, \ell, f)$ such that*

$$\left| \frac{1}{D} \sum_{d=1}^D L_d - \frac{1}{D} \sum_{d=1}^D L'_d \right| \leq \epsilon$$

*Additionally, for every $\phi \in \Phi$, there exists $(\phi', L') \in S^\epsilon(\tilde{X}, \tilde{Y}, \Phi, \ell, f)$ such that*

$$\left| \frac{1}{D} \sum_{d=1}^D \ell(f(\hat{x}^d, \phi), \hat{y}^d) - \frac{1}{D} \sum_{d=1}^D L'_d \right| \leq \epsilon$$

*Proof.* The first inequality follows from the same proof as in Lemma 3.1. For the second inequality, let $\phi'$ be the binary approximation to $\phi$, and $L'$ defined by $L'_d = \ell(f(\tilde{x}^d, \phi'), \tilde{y}^d)$. Since $\tilde{x}, \tilde{y}, \phi'$ are approximations to

$\hat{x}, \hat{y}, \phi$, by Lipschitzness we know that

$$|\ell(f(\hat{x}^d, \phi), \hat{y}^d) - L'_d| = |\ell(f(\hat{x}^d, \phi), \hat{y}^d) - \ell(f(\tilde{x}^d, \phi'), \tilde{y}^d)|$$
$$\leq 2\mathcal{L}\gamma = \epsilon$$

$\square$

**Corollary D.3.** *Let $(\hat{\phi}, \hat{L})$ defined as*

$$(\hat{\phi}, \hat{L}) \in \underset{\phi, L}{\operatorname{argmin}} \frac{1}{D} \sum_{d=1}^{D} L_d$$
$$s.t \, (\phi, L) \in S^\epsilon(\tilde{X}, \tilde{Y}, \Phi, \ell, f)$$

*Then*

$$\left| \frac{1}{D} \sum_{d=1}^{D} \ell(f(\hat{x}^d, \phi^*), \hat{y}^d) - \frac{1}{D} \sum_{d=1}^{D} \ell(f(\hat{x}^d, \hat{\phi}), \hat{y}^d) \right| \leq 2\epsilon$$

*Proof.* Since $\hat{\phi} \in \Phi$, and $\phi^*$ is a "true" optimal solution to the ERM problem, we immediately have

$$\frac{1}{D} \sum_{d=1}^{D} \ell(f(\hat{x}^d, \phi^*), \hat{y}^d) \leq \frac{1}{D} \sum_{d=1}^{D} \ell(f(\hat{x}^d, \hat{\phi}), \hat{y}^d)$$

On the other hand, by the previous Lemma we know there exists $(\phi', L') \in S^\epsilon(\tilde{X}, \tilde{Y}, \Phi, \ell, f)$ such that

$$-\epsilon \leq \frac{1}{D} \sum_{d=1}^{D} \ell(f(\hat{x}^d, \phi^*), \hat{y}^d) - \frac{1}{D} \sum_{d=1}^{D} L'_d$$

$$\leq \frac{1}{D} \sum_{d=1}^{D} \ell(f(\hat{x}^d, \phi^*), \hat{y}^d) - \frac{1}{D} \sum_{d=1}^{D} \hat{L}_d \qquad \text{(optimality of } \hat{L})$$

$$= \frac{1}{D} \sum_{d=1}^{D} \ell(f(\hat{x}^d, \phi^*), \hat{y}^d) - \frac{1}{D} \sum_{d=1}^{D} \ell(f(\tilde{x}^d, \hat{\phi}), \tilde{y}^d)$$

$$\leq \frac{1}{D} \sum_{d=1}^{D} \ell(f(\hat{x}^d, \phi^*), \hat{y}^d) - \frac{1}{D} \sum_{d=1}^{D} \ell(f(\hat{x}^d, \hat{\phi}), \hat{y}^d) + \epsilon \qquad \text{(Lipschitzness)}$$

$\square$

Note that since the objective is linear, the optimization problem in the previous Corollary is equivalent if we replace $S^\epsilon(\tilde{X}, \tilde{Y}, \Phi, \ell, f)$ by its convex hull. Therefore the only missing link to the face property of the data-independent polytope is the following:

**Lemma D.4.** $\operatorname{conv}(S^\epsilon(\tilde{X}, \tilde{Y}, \Phi, \ell, f))$ *is a face of* $\operatorname{conv}(S^\epsilon(D, \Phi, \ell, f))$.

*Proof.* The proof follows from simply fixing variables in the corresponding **LBO** that describes $\operatorname{conv}(S^\epsilon(D, \Phi, \ell, f))$. For every $d \in [D]$ and $v \in \mathcal{F}_d$, we simply need to make $\lambda_v = 0$ whenever the $(x, y)$ components of $v$ do not correspond to $\tilde{X}, \tilde{Y}$. We know this is well defined, since $\tilde{X}, \tilde{Y}$ are already discretized, thus there must be some $v \in \mathcal{F}_d$ corresponding to them.

The structure of the resulting LP is the same as **LBO**, so the fact that it is exactly $\operatorname{conv}(S^\epsilon(\tilde{X}, \tilde{Y}, \Phi, \ell, f))$ follows. The fact that it is a face of $\operatorname{conv}(S^\epsilon(D, \Phi, \ell, f))$ follows from the fact that the procedure simply fixed some inequalities to be tight. $\square$

## E    Regularized ERM

A common practice to avoid over-fitting is the inclusion of regularizer terms in (1). This leads to problems of the form

$$\min_{\phi \in \Phi} \frac{1}{D} \sum_{i=1}^{D} \ell(f(\hat{x}^i, \phi), \hat{y}^i) + \lambda R(\phi), \tag{9}$$

where $R(\cdot)$ is a function, typically a norm, and $\lambda > 0$ is a parameter to control the strength of the regularization. Regularization is generally used to promote generalization and discourage over-fitting of the obtained ERM solution. The reader might notice that our arguments in Section 3 regarding the epigraph reformulation of the ERM problem and the tree-decomposition of its intersection graph can be applied as well, since the regularizer term does not add any extra interaction between the data-dependent variables.

The previous analysis extends immediately to the case with regularizers after appropriate modification of the architecture Lipschitz constant $\mathcal{L}$ to include $R(\cdot)$.

**Definition E.1.** Consider a regularized ERM problem (9) with parameters $D, \Phi, \ell, f, R$, and $\lambda$. We define its *Architecture Lipschitz Constant* $\mathcal{L}(D, \Phi, \ell, f, R, \lambda)$ as

$$\mathcal{L}(D, \Phi, \ell, f, R, \lambda) \doteq \mathcal{L}_\infty(\ell(f(\cdot, \cdot), \cdot) + \lambda R(\cdot)) \tag{10}$$

over the domain $\mathcal{K} = [-1, 1]^n \times \Phi \times [-1, 1]^m$.

## F    ERM under Network Structure

So far we have considered general ERM problems exploiting only the structure of the ERM induced by the finite sum formulations. We will now study ERM under Network Structure, i.e., specifically ERM problems as they arise in the context of Neural Network training. We will see that in the case of Neural Networks, we can exploit the sparsity of the network itself to obtain better LP formulations of $\text{conv}(S^\epsilon(D, \Phi, \ell, f))$.

Suppose the network is defined by a graph $\mathcal{G}$, and recall that in this case, $\Phi \subseteq [-1, 1]^{E(\mathcal{G})}$. By using additional auxiliary variables $s$ representing the node computations and activations, we can describe $S(D, \Phi, \ell, f)$ in the following way:

$$\begin{aligned}
S(D, \Phi, \ell, f) = \Big\{ &(x^1, \ldots, x^D, y^1, \ldots, y^D, \phi, L) \in [-1, 1]^{(n+m)D} \times \Phi \times \mathbb{R}^D \ : \\
&L_d = \ell(s^{k,d}, y^d) \\
&s_v^{i,d} = a_v(g_v(s^{i-1,d}, \phi(\delta^+(v)))) \quad \forall v \in V_i, i \in [k] \\
&s^{0,d} = x^d \Big\}.
\end{aligned}$$

The only difference with our original description of $S(D, \Phi, \ell, f)$ in (5) is that we explicitly "store" node computations in variables $s$. These new variables will allow us to better use the structure of $\mathcal{G}$.

*Assumption* F.1. To apply our approach in this context we need to further assume $\Phi$ to be the class of Neural Networks with normalized coefficients and *bounded node computations*. This means that we restrict to the case when $s \in [-1, 1]^{|V(\mathcal{G})|D}$.

Under Assumption F.1 we can easily derive an analog description of $S^\epsilon(D, \Phi, \ell, f)$ using this node-based representation of $S^\epsilon(D, \Phi, \ell, f)$. In such description we also include a binary representation of the auxiliary variables $s$. Let $\Gamma$ be the intersection graph of such a formulation of $S^\epsilon(D, \Phi, \ell, f)$ and $\Gamma_\phi$ be the sub-graph of $\Gamma$ induced by variables $\phi$. Using a tree-decomposition $(T, Q)$ of $\Gamma_\phi$ we can construct a tree-decomposition of $\Gamma$ the following way:

1. We duplicate the decomposition $D$ times $(T^i, Q^i)_{i=1}^D$, where each $(T^i, Q^i)$ is a copy of $(T, Q)$.

2. We connect the trees $T^i$ in a way that the resulting graph is a tree (e.g., they can be simply concatenated one after the other).

3. To each bag $Q_t^i$ with $t \in T^i$ and $i \in [D]$, we add all the data-dependent variables $L_d$ and the binary variables associated with the discretization of $x^d$, $s^{\cdot,d}$, and $y^d$. This adds $N_\gamma(|V(\mathcal{G})| + n + m)$ additional variables to each bag, as there is only one variable $s$ per data point per vertex of $\mathcal{G}$.

It is not hard to see that this is a valid tree-decomposition of $\Gamma$, of size $|T| \cdot D$ —since the bags were duplicated $D$ times— and width $N_\gamma(tw(\Gamma_\phi) + |V(\mathcal{G})| + n + m)$.

We now turn to providing a bound to $tw(\Gamma_\phi)$. To this end we observe the following:

1. The architecture variables $\phi$ are associated to edges of $\mathcal{G}$. Moreover, two variables $\phi_e, \phi_f$, with $e, f \in E$ appear in a common constraint if and only if there is a vertex $v$ such that $e, f \in \delta^+(v)$.

2. This implies that $\Gamma_\phi$ is a sub-graph of the *line graph* of $\mathcal{G}$. Recall that the line graph of a graph $\mathcal{G}$ is obtained by creating a node for each edge of $\mathcal{G}$ and connecting two nodes whenever the respective edges share a common endpoint.

The treewidth of a line graph is related to the treewidth of the base graph (see Bienstock (1990); Calinescu et al. (1998); Atserias (2008); Harvey and Wood (2018)). More specifically, $tw(\Gamma_\phi) \in O(tw(\mathcal{G})\Delta(\mathcal{G}))$ where $\Delta$ denotes the maximum vertex degree. Additionally, using Lemma A.2 we may assume $|T| \leq |E(\mathcal{G})|$, since $\Gamma_\phi$ has at most $|E(\mathcal{G})|$ nodes. Putting everything together we obtain:

**Lemma F.2.** *If there is an underlying network structure $\mathcal{G}$ in the ERM problem and the node computations are bounded, then* $\text{conv}(S^\epsilon(D, \Phi, \ell, f))$ *admits a linear programming formulation with no more than*

$$2D(|E(\mathcal{G})|+1)\left(\frac{2\mathcal{L}}{\epsilon}\right)^{O(tw(\mathcal{G})\Delta(\mathcal{G})+|V(\mathcal{G})|+n+m)} \quad and \quad D(|E(\mathcal{G})|+1)\left(2\left(\frac{2\mathcal{L}}{\epsilon}\right)^{O(tw(\mathcal{G})\Delta(\mathcal{G})+|V(\mathcal{G})|+n+m)} + 1\right)$$

*variables and constraints, respectively. Moreover, given a tree-decomposition of the network $\mathcal{G}$, the linear program can be constructed in time* $O\left(D|E(\mathcal{G})|(2\mathcal{L}/\epsilon)^{O(tw(\mathcal{G})\Delta(\mathcal{G})+|V(\mathcal{G})|+n+m)}\right)$ *plus the time required for* $O\left(|E(\mathcal{G})|(2\mathcal{L}/\epsilon)^{O(tw(\mathcal{G})\Delta(\mathcal{G})+|V(\mathcal{G})|+n+m)}\right)$ *evaluations of $\ell$ and $f$.*

## G  EXPLICIT LIPSCHITZ CONSTANTS OF COMMON LOSS FUNCTIONS

In Section 4 we specified our results —the size of the data-independent LPs— for feed-forward networks with 1-Lipschitz activation functions. However, we kept as a parameter $\mathcal{L}_\infty(\ell)$; the Lipschitz constant of $\ell(\cdot, \cdot)$ over $[-U_k, U_k]^m \times [-1, 1]^m$, with $U_k = \sum_{j=0}^k w^j$ a valid bound on the output of the node computations, as proved in Lemma 4.1. Note that $U_k \leq w^{k+1}$.

In this Section we compute this Lipschitz constant for various common loss functions. It is important to mention that we are interested in the Lipschitznes of $\ell$ with respect to both the output layer and the data-dependent variables as well —not a usual consideration in the literature. Note that a bound on the Lipschitz constant $\mathcal{L}_\infty(\ell)$ is given by $\sup_{z,y} \|\nabla\ell(z, y)\|_1$.

- Quadratic Loss $\ell(z, y) = \|z - y\|_2^2$. In this case it is easy to see that

$$\|\nabla\ell(z, y)\|_1 = 4\|z - y\|_1 \leq 4m(U_k + 1) \leq 4m(w^{k+1} + 1)$$

- Absolute Loss $\ell(z, y) = \|z - y\|_1$. In this case we can directly verify that the Lipschitz constant with respect to the infinity norm is at most $2m$.

- Cross Entropy Loss with Soft-max Layer. In this case we include the Soft-max computation in the definition of $\ell$, therefore

$$\ell(z, y) = -\sum_{i=1}^{m} y_i \log(S(z)_i)$$

where $S(z)$ is the Soft-max function defined as

$$S(z)_i = \frac{e^{z_i}}{\sum_{j=1}^{m} e^{z_j}}.$$

A folklore result is

$$\frac{\partial \ell(z, y)}{\partial z_i} = S(z)_i - y_i \Rightarrow \left| \frac{\partial \ell(z, y)}{\partial z_i} \right| \leq 2$$

Additionally,

$$\frac{\partial \ell(z, y)}{\partial y_i} = -\log(S(z)_i)$$

which in principle cannot be bounded. Nonetheless, since we are interested in the domain $[-U_k, U_k]$ of $z$, we obtain

$$S(z)_i = \frac{e^{z_i}}{\sum_{j=1}^{m} e^{z_j}} \geq \frac{1}{m} e^{-2U_k} \Rightarrow \left| \frac{\partial \ell(z, y)}{\partial y_i} \right| = -\log(S(z)_i) \leq \log(m) + 2U_k$$

which implies that $\mathcal{L}_\infty(\ell) \leq 2m(\log(m) + 2U_k) \leq 2m(\log(m) + 2w^{k+1})$.

- Hinge Loss $\ell(z, y) = \max\{1 - z^T x, 0\}$. Using a similar argument as for the Quadratic Loss, one can easily see that the Lipschitz constant with respect to the infinity norm is at most $m(U_k + 1) \leq m(w^{k+1} + 1)$.

## H  BINARIZED NEURAL NETWORKS

A *Binarized activation unit (BiU)* is parametrized by $p + 1$ values $b, a_1, \ldots, a_p$. Upon a binary input vector $z_1, z_2, \ldots, z_p$ the output is binary value $y$ defined by:

$$y = 1 \text{ if } a^T z > b, \quad \text{and } y = 0 \text{ otherwise.}$$

Now suppose we form a network using BiUs, possibly using different values for the parameter $p$. In terms of the training problem we have a family of (binary) vectors $x^1, \ldots, x^D$ in $\mathbb{R}^n$ and binary labels and corresponding binary label vectors $y^1, \ldots, y^D$ in $\mathbb{R}^m$, and as before we want to solve the ERM problem (1). Here, the parametrization $\phi$ refers to a choice for the pair $(a, b)$ at each unit. In the specific case of a network with 2 nodes in the first layer and 1 node in the second layer, and $m = 1$, Blum and Rivest (1992) showed that it is NP-hard to train the network so as to obtain zero loss, when $n = D$. Moreover, the authors argued that even if the parameters $(a, b)$ are restricted to be in $\{-1, 1\}$, the problem remains NP-Hard. See Courbariaux et al. (2016) for an empirically efficient training algorithm for BiUs.

In this section we apply our techniques to the ERM problem (1) to obtain an *exact* polynomial-size *data-independent formulation* for each fixed network (but arbitrary $D$) when the parameters $(a, b)$ are restricted to be in $\{-1, 1\}$.

We begin by noticing that we can reformulate (1) using an epigraph formulation as in (4). Moreover, since the data points in a BiU are binary, if we keep the data points as variables, the resulting linear-objective optimization problem is a binary optimization problem as **BO**. This allows us to claim the following:

**Theorem H.1.** *Consider a graph $\mathcal{G}$, $p \in \mathbb{N}$ and $D \in \mathbb{N}$. There exists a linear program of size*

$$O(2^{p|V(\mathcal{G})|}D),$$

*such that any BiU ERM problem of the form (1) is equivalent to optimizing a linear function over a face of P, where P is the polytope defined by the linear program. Constructing the linear program takes time $O(2^{p|V(\mathcal{G})|}D)$ plus the time required for $O(2^{p|V(\mathcal{G})|})$ evaluations of $f$ and $\ell$.*

*Proof.* The result follows from applying Theorem 2.5 directly to the epigraph formulation of BiU keeping $x$ and $y$ as variables. In this case an approximation is not necessary. The construction time and the data-independence follow along the same arguments used in the approximate setting before. □

The following corollary is immediate.

**Corollary H.2.** *The ERM problem (1) over BiUs can be solved in polynomial time for any D, whenever $p$ and the network structure $\mathcal{G}$ are fixed.*

## I    LINEAR PROGRAMMING-BASED TRAINING GENERALIZES

In this section we will show that the ERM solutions obtained via Linear Programming generalize to the General Risk Minimization problem. Here we show generalization as customary in stochastic optimization, exploiting the Lipschitzness of the model to be trained; we refer the interested reader to Ahmed (2017); Shapiro et al. (2009) for an in-depth discussion. In a first step, we further precise notation as required for our analysis. To this end, recall that the *General Risk Minimization (GRM)* is defined as $\min_{\phi \in \Phi} \mathrm{GRM}(\phi) \doteq \min_{\phi \in \Phi} \mathbb{E}_{(x,y) \in \mathcal{D}} [\ell(f(x,\phi), y)]$, where $\ell$ is some *loss function*, $f$ is a neural network architecture with parameter space $\Phi$, and $(x, y) \in \mathbb{R}^{n+m}$ drawn from the distribution $\mathcal{D}$. We solve the finite sum problem, i.e., the *empirical risk minimization problem* $\min_{\phi \in \Phi} \mathrm{ERM}_{X,Y}(\phi) \doteq \min_{\phi \in \Phi} \frac{1}{D} \sum_{i=1}^{D} \ell(f(x^i, \phi), y^i)$, instead, where $(X, Y) = (x^i, y^i)_{i=1}^{D}$ is an i.i.d. sample from data distribution $\mathcal{D}$ of size $D$. We will show in this section, for any $1 > \alpha > 0$, $\epsilon > 0$, we can choose a (reasonably small!) sample size $D$, so that with probability $1 - \alpha$ it holds:

$$\mathrm{GRM}(\bar{\phi}) \leq \min_{\phi \in \Phi} \mathrm{GRM}(\phi) + 6\epsilon,$$

where $\bar{\phi} \leq \max_{\phi \in \Phi} \mathrm{ERM}_{X,Y}(\phi) + \epsilon$ is an $\epsilon$-approximate solution to $\mathrm{ERM}_{X,Y}$ for i.i.d.-sampled data $(X, Y) \sim \mathcal{D}$. As the size of the linear program that we use for training only *linearly* depends on the number of data points, this also implies that we will have a linear program of reasonable size as a function of $\alpha$ and $\epsilon$.

The following proposition summaries the generalization argument used in stochastic programming as presented in Ahmed (2017) (see also Shapiro et al. (2009)):

**Proposition I.1.** *Consider the optimization problem*

$$\min_{x \in X} \mathbb{E}_{\omega \in \Omega} [F(x, \gamma(\omega))],$$

*where $\gamma(\omega)$ is a random parameter with $\omega \in \Omega$ a set of parameters, $X \subseteq \mathbb{R}^n$ a finite set, and $F : X \times \Omega \to \mathbb{R}$ is a function. Given i.i.d. samples $\gamma_1, \ldots, \gamma_D$ of $\gamma(\omega)$, consider the finite sum problem*

$$\min_{x \in X} \frac{1}{D} \sum_{i \in [D]} F(x, \gamma_i). \tag{11}$$

*If $\bar{x} \in X$ is an $\epsilon$-approximate solution to (11), i.e., $\frac{1}{D} \sum_{i \in [D]} F(\bar{x}, \gamma_i) \leq \min_{x \in X} \frac{1}{D} \sum_{i \in [D]} F(x, \gamma_i) + \epsilon$ and*

$$D \geq \frac{4\sigma^2}{\epsilon^2} \log \frac{|X|}{\alpha},$$

where $\alpha > 0$ and $\sigma^2 = \max_{x \in X} \mathbb{V}_{\omega \in \Omega} [F(x, \gamma(\omega))]$, then with probability $1 - \alpha$ it holds:

$$\mathbb{E}_{\omega \in \Omega} [F(\bar{x}, \gamma(\omega))] \leq \min_{x \in X} \mathbb{E}_{\omega \in \Omega} [F(x, \gamma(\omega))] + 2\epsilon.$$

Let $\sigma^2 = \max_{\phi \in \Phi} \mathbb{V}_{(x,y) \in \mathcal{D}} [\ell(f(x, \phi), y)]$. We will now establish generalization by means of Proposition I.1 and a straightforward discretization argument. By assumption from above $\Phi \subseteq [-1, 1]^N$ for some $N \in \mathbb{N}$. Let $\Phi_\nu \subseteq \Phi \subseteq [-1, 1]^N$ be a $\nu$-net of $\Phi$, i.e., for all $\phi \in \Phi$ there exists $\bar{\phi} \in \Phi_\nu$ with $\|\phi - \bar{\phi}\|_\infty \leq \nu$. Furthermore let $\mathcal{L}$ the be architecture Lipschitz constant, as defined in (2) (or (10)).

**Theorem I.2.** *[Generalization] Let $\bar{\phi} \in \Phi$ be an $\epsilon$-approximate solution to $\min_{\phi \in \Phi} \mathrm{ERM}_{X,Y}(\phi)$ with $\epsilon > 0$, i.e., $\mathrm{ERM}_{X,Y}(\bar{\phi}) \leq \min_{\phi \in \Phi} \mathrm{ERM}_{X,Y}(\phi) + \epsilon$. If $D \geq \frac{4\sigma^2}{\epsilon^2} \log \frac{((2\mathcal{L})/\epsilon)^N}{\alpha}$, with $\mathcal{L}$ and $\sigma^2$ as above, then with probability $1 - \alpha$ it holds $\mathrm{GRM}(\bar{\phi}) \leq \min_{\phi \in \Phi} \mathrm{GRM}(\phi) + 6\epsilon$, i.e., $\bar{\phi}$ is a $6\epsilon$-approximate solution to $\min_{\phi \in \Phi} \mathrm{GRM}(\phi)$.*

*Proof.* Let $\bar{\phi}$ be as above. With the choice $\nu \doteq \epsilon/\mathcal{L}$, there exists $\tilde{\phi} \in \Phi_\nu$, so that $\|\tilde{\phi} - \bar{\phi}\|_\infty \leq \nu$ and hence by Lipschitzness,

$$|\mathrm{ERM}_{X,Y}(\bar{\phi}) - \mathrm{ERM}_{X,Y}(\tilde{\phi})| \leq \epsilon,$$

so that $\mathrm{ERM}_{X,Y}(\tilde{\phi}) \leq \min_{\phi \in \Phi_\nu} \mathrm{ERM}_{X,Y}(\phi) + 2\epsilon$. As $D \geq \frac{4\sigma^2}{\epsilon^2} \log \frac{((2\mathcal{L})/\epsilon)^N}{\alpha}$, with probability $1 - \alpha$ we have $\mathrm{GRM}(\tilde{\phi}) \leq \min_{\phi \in \Phi_\nu} \mathrm{GRM}(\phi) + 4\epsilon$ by Proposition I.1. If now $\bar{\phi}_G = \mathrm{argmin}_{\phi \in \Phi} \mathrm{GRM}(\phi)$ and $\tilde{\phi}_G \in \Phi_\nu$ with $\|\bar{\phi}_G - \tilde{\phi}_G\|_\infty \leq \nu$, by Lipschitzness we have $|\mathrm{GRM}(\bar{\phi}_G) - \mathrm{GRM}(\tilde{\phi}_G)| \leq \epsilon$. Now

$$\mathrm{GRM}(\tilde{\phi}) \leq \min_{\phi \in \Phi_\nu} \mathrm{GRM}(\phi) + 4\epsilon$$

$$\leq \mathrm{GRM}(\tilde{\phi}_G) + 4\epsilon \qquad \text{(by optimality)}$$

$$\leq \mathrm{GRM}(\bar{\phi}_G) + 5\epsilon \qquad \text{(by Lipschitzness)}.$$

Together with $|\mathrm{GRM}(\bar{\phi}) - \mathrm{GRM}(\tilde{\phi})| \leq \epsilon$ as $\|\tilde{\phi} - \bar{\phi}\|_\infty \leq \nu$ it follows

$$\mathrm{GRM}(\bar{\phi}) \leq \mathrm{GRM}(\bar{\phi}_G) + 6\epsilon = \min_{\phi \in \Phi} \mathrm{GRM}(\phi) + 6\epsilon,$$

which completes the proof. $\qquad\square$

We are ready to formulate the following corollary combining Theorem I.2 and Main Theorem 3.1.

**Corollary I.3** (LP-based Training for General Risk Minimization). *Let $\mathcal{D}$ be a data distribution as above. Further, let $1 > \alpha > 0$ and $\epsilon > 0$, then there exists a linear program with the following properties:*

(a) *The linear program has size $O\left(\left(\frac{2\mathcal{L}}{\epsilon}\right)^{n+m+N} \left(\frac{4\sigma^2}{\epsilon^2} \log \frac{(2\mathcal{L}/\epsilon)^N}{\alpha}\right)\right)$ and can be constructed in time $O\left((2\mathcal{L}/\epsilon)^{n+m+N} \left(\frac{4\sigma^2}{\epsilon^2} \log \frac{((2\mathcal{L})/\epsilon)^N}{\alpha}\right)\right)$ plus the time required for $O\left((2\mathcal{L}/\epsilon)^{n+m+N}\right)$ evaluations of $\ell$ and $f$, where $\mathcal{L}$ and $\sigma^2$ as above.*

(b) *With probability $(1 - \alpha)$ it holds $\mathrm{GRM}(\bar{\phi}) \leq \min_{\phi \in \Phi} \mathrm{GRM}(\phi) + 6\epsilon$, where $\bar{\phi}$ is an optimal solution to the linear program obtained for the respective sample of $\mathcal{D}$ of size $\frac{4\sigma^2}{\epsilon^2} \log \frac{((2\mathcal{L})/\epsilon)^N}{\alpha}$.*

Similar corollaries hold, combining Theorem I.2 with the respective alternative statements from Section 4. Of particular interest for what follows is the LP size in the case of a neural network with $k$ layers with width $w$, which becomes

$$O\left(\left(2\mathcal{L}_\infty(\ell)w^{O(k^2)}/\epsilon\right)^{n+m+N} \left(4\sigma^2/\epsilon^2\right) \log((2\mathcal{L}_\infty(\ell)w^{O(k^2)}/\epsilon)^N/\alpha)\right). \tag{12}$$

A closely related result regarding an approximation to the GRM problem for neural networks is provided by Goel et al. (2017) in the improper learning setting. The following corollary to (Goel et al., 2017, Corollary 4.5) can be directly obtained, rephrased to match our notation:

**Theorem I.4** (Goel et al. (2017))**.** *There exists an algorithm that outputs $\tilde{\phi}$ such that with probability $1 - \alpha$, for any distribution $\mathcal{D}$ and loss function $\ell$ which is convex, L-Lipschitz in the first argument and b bounded on $[-2\sqrt{w}, \sqrt{w}]$, $\mathrm{GRM}(\tilde{\phi}) \leq \min_{\phi \in \Phi} \mathrm{GRM}(\phi) + \epsilon$, where $\Phi$ is the class of neural networks with k hidden layers, width w, output dimension $m = 1$, ReLU activations and normalized weights. The algorithm runs in time at most*

$$n^{O(1)} 2^{((L+1)w^{k/2}k\epsilon^{-1})^k} \log(1/\alpha) \tag{13}$$

*Remark* I.5. In contrast to the above result of Goel et al. (2017), note that in our paper we consider the proper learning setting, where we actually obtain a neural network. In addition we point out several key differences between Theorem I.4 and the algorithmic version of our result when solving the LP in Corollary I.3 of size as (12): (a) In (13), the dependency on the input dimension is better than in (12). (b) The dependency on the Lipschitz constant is significantly better in (12), although we have to point out that we are relying on the Lipschitz constant with respect to *all* inputs of the loss function and in a potentially larger domain. (c) The dependency on $\epsilon$ is also better in (12). (d) We are not assuming convexity of $\ell$ and we consider general $m$. (e) The dependency on $k$ in (12) is much more benign than the one in (13), which is doubly exponential.

*Remark* I.6. Since the first submission of this article, a manuscript by Manurangsi and Reichman (2018) was published which extended the results by Goel et al. (2017). This work provides an algorithm with similar characteristics to the one by Goel et al. (2017) but in the *proper* learning setting, for depth-2 ReLU networks with convex loss functions. The running time of the algorithm (rephrased to match our notation) is $(n/\alpha)^{O(1)} 2^{(w/\epsilon)^{O(1)}}$. Analogous to the comparison in Remark I.5, we obtain a much better dependence with respect to $\epsilon$ and we do not rely on convexity of the loss function or on constant depth of the neural network.

