# OpenReview forum: "Principled Deep Neural Network Training through Linear Programming"
_ICLR.cc/2019/Conference_

### Official Review · AnonReviewer1 · 2018-11-02
**Solid work that could discuss the implications for the ICLR community better**

**Rating:** 8
**Confidence:** 3

**Review:**

This is very solid work and the framework allows one to plug-in existing complexity measures to provide complexity upper bounds for (some) DNNs. The main idea is to rephrase an empirical risk minimization problem in terms of a binary optimization problem
using a discretization of the continuous variables. Then this formulation is used to provide a as a moderate-sized linear program of its convex hull.

In my opinion, every paper that provides insights into the complexity and generalization of deep learning is an important contribution. Moreover, the present paper is based on
a recent insight of the authors, i.e., it is based on solid grounds. However, it would have been nice to also show some practical insights. The main take-aways message is that we need exponential time. Is this practical for networks with with millions of parameters? Or does this imply that deep learning is hopeless (in theory)? To be fair, the authors touch upon this in the conclusions, but only 1-2 sentences. This discussion should be extended. Nevertheless, I agree that the bridge built is important and may indeed trigger some very important future contributions.

The authors should, however, also review other work on linear programming for deep networks coming from the machine learning community such as

Brandon Amos, Lei Xu, J. Zico Kolter:
Input Convex Neural Networks.
ICML 2017: 146-155

Given the background of the average ICLR reader, the authors should also introduce (at least the intuitions) improper and proper learning setups in the introduction before using them.   This also holds for other terminology from complexity theory. Indeed, the authors cannot introduce/review all complexity theory. However, they should try their best and fill the rest by a reference to an introductionary book or directly to the appendic. Without, while important for the ICLR community, the authors run the risk that the paper would better be suited by a learning theory venue.

---

> ### Author Response · Authors · 2018-11-13
> **Review response**
>
> First of all, thank you very much for the thorough review. We are glad you find our work a solid contribution. We are currently working on an improved version of the paper, which will include your suggestions and clarify your concerns. In the meantime, we would like to comment on all points raised in your review.
>
> - "It would have been nice to also show some practical insights. The main take-aways message is that we need exponential time. Is this practical for networks with with millions of parameters? Or does this imply that deep learning is hopeless (in theory)? To be fair, the authors touch upon this in the conclusions, but only 1-2 sentences. This discussion should be extended. Nevertheless, I agree that the bridge built is important and may indeed trigger some very important future contributions."
>
> Thank you very much for viewing our work as providing a bridge between communities; this was indeed our intention. We will extend the discussion on the exponential dependency of our approach. Regarding the practicality of our approach, given that the three reviewers touched on this aspect we provide a justification and clarification in a separate message in this forum for the three referees. We hope you find it satisfactory.
>
> - "The authors should, however, also review other work on linear programming for deep networks coming from the machine learning community such as Brandon Amos, Lei Xu, J. Zico Kolter: Input Convex Neural Networks. ICML 2017: 146-155"
>
> Thank you very much for the pointer to this work. We will review this work and add the corresponding citation in the revised manuscript.
>
> - "Given the background of the average ICLR reader, the authors should also introduce (at least the intuitions) improper and proper learning setups in the introduction before using them.   This also holds for other terminology from complexity theory. Indeed, the authors cannot introduce/review all complexity theory. However, they should try their best and fill the rest by a reference to an introductionary book or directly to the appendic. Without, while important for the ICLR community, the authors run the risk that the paper would better be suited by a learning theory venue."
>
> Thank you for this suggestion. We will provide a more careful introduction for technical terminology to expand the reach of our paper.

---

### Official Review · AnonReviewer2 · 2018-11-02

**Rating:** 6
**Confidence:** 3

**Review:**

This paper studies the problem of proper learning of deep neural network. In particular, the focus is on doing
approximate empirical risk minimization over the class of neural networks of a fixed architecture. The main
result of the paper is that approximate ERM can be formulated as an LP problem that is of size exponential in the
network parameters and the input dimensionality. The paper uses a framework of Bienstock and Munoz that shows how to
write a binary optimization problem as a linear problem with size dependent on the treewidth of an appropriate graph
associated with the optimization problem. In order to apply the framework, the authors first discretize the parameter
space appropriately and then apply analyze the treewidth of the discretized space. The authors also provide treewidth
analysis of specific architectures including fully connected networks, and CNNs with various activations.

Most of the technical work in the paper involves analyzing the treewidth of the resulting discretized problem. The nice
feature of the result is that it holds for worst case data sets, and hence, the exponential dependence on various
parameters is unavoidable. On the other hand, it is unclear to me as to how these ideas might eventually lead to
practical algorithms or shed light on current training practices in the deep learning community. For instance, it would
be very interesting to investigate if under certain assumptions on the data generation process, one can get small LPs
that depend exponentially only in the depth, as opposed to the input dimensionality.

I also feel that section 5 does not add much to the main results of the paper and can be skipped or moved entirely to the appendix. On a technical note, I don't see where the dependence on the input dimensionality appears in Theorem 5.1.

---

> ### Author Response · Authors · 2018-11-13
> **Review response**
>
> First of all, thank you very much for the thorough review. We appreciate the feedback provided and we are happy that you consider the "worst-case" feature of our approach an important one. We are currently working on an improved version of the paper, which will include your suggestions and clarify your concerns. In the meantime, we would like to comment on the points raised in this review.
>
> - "It is unclear to me as to how these ideas might eventually lead to practical algorithms or shed light on current training practices in the deep learning community."
>
> Given that the three reviewers touched on this aspect we provide a justification and clarification in a separate message in this forum. We hope you find it satisfactory.
>
> - "For instance, it would be very interesting to investigate if under certain assumptions on the data generation process, one can get small LPs that depend exponentially only in the depth, as opposed to the input dimensionality."
>
> The reviewer is correct. In order to provide a cleaner analysis, we are assuming that all data in "[-1,1]^{n+m}" is a possible input. The "n+m" term in the exponent of the LP sizes is a consequence of this assumption as per our approach. However, as the reviewer noted, one can make use of additional structure in the data generation process in order to alleviate the LP size from the input-dimensionality.
>
> If each data point belongs to a set "U", for example, and our grid over "U" has "M" points then our LP size will depend on "M" instead of "(2L/\epsilon)^{n+m}". Only the parameter space dimension will remain in the exponent, as the reviewers suggests.
>
> While interesting in its own right, this is beyond the scope of the current paper and left for future work. However, we will add a remark on this in the new manuscript for the curious reader.
>
> - "I also feel that section 5 does not add much to the main results of the paper and can be skipped or moved entirely to the appendix. On a technical note, I don't see where the dependence on the input dimensionality appears in Theorem 5.1."
>
> We believe Section 5 is necessary, as Generalization of ERM estimators is an important feature to have. Nonetheless, we agree with the reviewer in that it is not at the core of the main results. The new version will have the entire Generalization discussion in the Appendix.
>
> Regarding the dependence on the input dimensionality, it is correct that Theorem 5.1 does not need any. We are working with Lipschitz constants that depend on the infinity norm, which only looks at entries individually. Moreover, this Theorem is only making a statement on the minimum number of data points needed to achieve a certain approximation guarantee, which depends on the distribution of the data (through  "\sigma^2") but not necessarily on their dimension.

---

### Official Review · AnonReviewer3 · 2018-11-04
**exponential complexity; practical relevance unclear**

**Rating:** 6
**Confidence:** 4

**Review:**

This work reformulates the neural network training as an LP with size that is exponential in the size of the architecture and data dimension, and polynomial in the size of the data set. They further analyze generalization properties. It extends previous works on 1-hidden-layer neural-nets (say, In Arora et al. (2018)).

Pros: Establish new time complexity (to my knowledge) of general neural-nets.

Cons: It seems far from having a practical implication. Exponential complexity is huge (though common in TCS and IP communities). No simulation was presented. Not sure which part of the approach is useful for practitioners.
    My feeling is that the paper is a bit too theoretical and less relevant to ICLR audience. More theoretical venues may be a better fit.

Other questions:
--The authors mentioned “that is exponential in the size of the architecture (and data dimension)  and polynomial in the size of the data set;” and "this is the best one can hope for due to the NP-Hardness of the problem ".
a)	The time complexity is exponential in both the size of neural-net and the data dimension (the latter seems to be ignored in abstract). Is there a reference that presents results on NP-hardness in terms of both parameters, or just one parameter?
b)	The NP-hardness reduction may give an exp. time algorithm. Is there a simple exponential time algorithm? If so, I expect the dependence on the size of the data set is exponential, and the contribution of this paper is to improve to polynomial. The authors mentioned one discretization method, but are there others? More explanation of the importance of the proved time complexity will be helpful.

-- Novelty in technical parts: The idea of tree-width graph was introduced in Bienstock and Muñoz (2018). The main theorem 3.1 is based on explicit construction for Theorem 2.5, and Theorem 2.5 is an immediate generalization of a theorem in Bienstock and Muñoz (2018) as mentioned in the paper. Thus, this paper looks like an easy extension of Bienstock and Muñoz (2018) --intuitively, minimizing polynomials by LP seems to be closely related to solving neural-nets problems by LP. Could the authors explain more on the technical novelty?

Update after rebuttal: I'd like to thank the authors for the detailed response. It addressed most of my concerns.
    The analogy with MIP makes some sense, as huge LPs are indeed being solved every day. However, an important difference is that those problems cannot be solved in other better ways till now, while for deep learning people are already successfully solving the current formulations. I still think this work will probably not lead to a major empirical improvement.
     I just realize that my concern on the practical relevance is largely due to the title "Principled Deep Neural Network Training through Linear Programming". It sounds like it can provide a better "training" method, but it does not show a practical algorithm that works for CIFAR10 at this stage. The title should not sound like "look, here is a new method that can change training", but "hey, check some new theoretical progress, it may lead to future progress". I strongly suggest changing the title to something like "Reformulating DNN as a uniform LP" or "polynomial time algorithm in the input dimension", which reflects the theoretical nature.
    That being said, the MIP analogy makes me think that there might be some chance that this LP formulation is useful in the future, maybe for solving some special problems that current methods fail miserably.  In addition, it does provide a solid theoretical contribution. For those reasons (and assuming the title will be changed), I increase my score.

---

> ### Author Response · Authors · 2018-11-13
> **Review response (1)**
>
> We would like to thank the reviewer for the comments and suggestions. The reviewer raised some important points that we hope the next version will amend.  We are currently working on an improved version of the paper that will include these. In the meantime, we would like to comment on all points raised by this review.
>
> - "It seems far from having a practical implication. Exponential complexity is huge (though common in TCS and IP communities). No simulation was presented. Not sure which part of the approach is useful for practitioners. My feeling is that the paper is a bit too theoretical and less relevant to ICLR audience. More theoretical venues may be a better fit."
>
> We have provided a justification and clarification in an above statement in this forum. We hope you find it satisfactory.
>
> - "The authors mentioned “that is exponential in the size of the architecture (and data dimension)  and polynomial in the size of the data set;” and "this is the best one can hope for due to the NP-Hardness of the problem ".  a) The time complexity is exponential in both the size of neural-net and the data dimension (the latter seems to be ignored in abstract). Is there a reference that presents results on NP-hardness in terms of both parameters, or just one parameter?"
>
> The referee raises an interesting point regarding whether these two parameters can be decoupled. The input dimension and the parameter space dimension are typically related to each other in the NP-hardness results. For example, in the paper by Blum and Rivest (1992), NP-Hardness of the training problem is proved with respect to a parameter "n" which is the input dimension *and* the parameter space dimension (roughly). If we plug in that architecture size into our result, we obtain an exponential dependency on n. In a sense, "the best one can hope for".
>
> A recent paper submitted to this conference, entitled "Complexity of Training ReLU Neural Network", works on this subject as well. They prove polynomial time solvability of the training problem for fixed input dimension, however, they consider a *fixed* architecture (the polynomial dependency is with respect to the sample size).
>
> If these parameters are decoupled, it is not known if the exponential dependence in the parameter space dimension can be alleviated. Quoting the paper by Arora et al. (2018):
>
> "we are not aware of any complexity results which would rule out the possibility of an algorithm which trains to global optimality in time that is polynomial in the data size and/or the number of hidden nodes, assuming that the input dimension is a fixed constant"
>
> As the referee noted, our phrasing is a bit confusing. In the phrase quoted by the reviewer, we were considering all the term "n+m+N" to be the "size of the architecture", since the input dimension can be also considered as part of the architecture. We now realize this is not standard, and somewhat confusing, so we will clarify this in the revised version.

---

> ### Author Response · Authors · 2018-11-13
> **Reviewer response (2)**
>
> - "b) The NP-hardness reduction may give an exp. time algorithm. Is there a simple exponential time algorithm? If so, I expect the dependence on the size of the data set is exponential, and the contribution of this paper is to improve to polynomial."
>
> Recall that our result constructs a *uniform* LP, meaning, an LP that encodes (approximately) all input data-sets and that can be used for training with any input data. Constructing such LP is also NP-hard since it includes the NP-hard setting (as a face).
>
> As the reviewer points out, there is a simple exponential time algorithm that can do the same: simply consider an epsilon-grid to enumerate over all possible (approximate) inputs, and for each input solve the training problem. This would yield an algorithm with exponential dependency on the sample size.
>
> In the non-uniform case, however, one can achieve a polynomial dependency on the sample size with a simple algorithm. We refer the Reviewer to Appendix B of the current version, where we discuss the differences between a "uniform" and a "non-uniform" LP.
>
> The contribution of this paper, we believe, is not only that we provide a linear dependency on the data, but more importantly that we show that one can give a uniform LP (of size linear in D) that would work for all samples of a given size, without compromising on the linear dependence on the sampling size.
>
> - "The authors mentioned one discretization method, but are there others? More explanation of the importance of the proved time complexity will be helpful."
>
> Regarding the importance of the proved time complexity, we address this comment in the common response to all reviewers in this forum.
>
> Regarding the discretization method, the reviewer touches on an important question in non-convex optimization and algebraic geometry, and an active research field. For example, the work of Piazzon et al. [1] or Vianello [2], study how one can discretize a potentially non-convex region while satisfying certain approximation guarantees. In general, this is a complex task when little structure is present, as we are assuming in order to tackle the general ERM problem. If there is a stronger geometrical structure, however, one can discretize in a more efficient way. A classical example is the Ben-Tal and Nemirovski [3] approximation of the second-order cone, where a custom discretization is performed which uses the geometry of such convex cone. Nonetheless, the non-convexities present in the DNN setting made us stick to the "inverse powers of 2" discretization, as it is clean and provides an efficient approximation.
>
> [1] Piazzon, Federico, and Marco Vianello. "Jacobi norming meshes." Math. Inequal. Appl 19 (2016): 1089-1095.
> [2] Vianello, Marco. "Norming meshes by Bernstein-like inequalities." Math. Inequal. Appl 17.3 (2014): 929-936.
> [3] Ben-Tal, Aharon, and Arkadi Nemirovski. "On polyhedral approximations of the second-order cone." Mathematics of Operations Research 26.2 (2001): 193-205.

---

> ### Author Response · Authors · 2018-11-21
> **Reviewer response (3)**
>
> Dear Reviewer,
>
> The last part of our response was not posted at the same time as the others by mistake. Please find it below:
>
> - "Novelty in technical parts: the idea of tree-width graph was introduced in Bienstock and Munoz (2018). The main theorem 3.1 is based on explicit construction for Theorem 2.5, and Theorem 2.5 is an immediate generalization of a theorem in Bienstock and Munoz (2018) as mentioned in the paper. Thus, this paper looks like an easy extension of Bienstock and Munoz (2018) --intuitively, minimizing polynomials by LP seems to be closely related to solving neural-nets problems by LP. Could the authors explain more on the technical novelty? "
>
> As the referee points out, the main results rely on ideas developed in [1]. However, the results in this submission do not follow as direct corollaries, as non-trivial extensions are needed to be made in [1] before applying the result to the ERM problem:
>
>    - In [1] only polynomials are considered, whose domain is "[0,1]". For this paper we needed to extend the results beyond polynomials, as many DNN architectures present non-differentiable structures. Indeed, the discussions involving Lipschitz constants are all new.
>    - We also needed to extend the result to allow the presence of some unbounded variables, which are the output of the loss function.
>
> These extensions are key ingredients that allow us to use [1] in this new ERM setting. In addition, the fact that the ERM problem can be formulated as an optimization problem whose treewidth does not depend on the data is a novel an potentially impactful result. This is what allows us to use [1] (which in contrast assumes a certain treewidth is given), and will hopefully trigger other important contributions.
>
> We think that the most surprising technical novelty concerns the *uniform* LP, i.e., an LP that can be written before seeing the data. This result can be viewed as follows: even though the set of all training sets is infinite,  this set can be partitioned into a finite set of classes (for a given architecture) and each class corresponds to the face of a polyhedron -- and the solution to our LP on that face provides a provably (nearly) optimal solution to the training problem.  Moreover, the size of the LP (variables and constraints) is linear in D. These results are stronger than previous results and answer several open questions left by previous work.
>
> [1] Bienstock, Daniel, and Gonzalo Munoz. "LP Formulations for Polynomial Optimization Problems." SIAM Journal on Optimization 28.2 (2018): 1121-1150.

---

> ### Author Response · Authors · 2018-11-29
> **thank you for the update**
>
> We are glad our replies addressed most concerns of the reviewer and we are thankful for the score increase. We agree with the reviewer in that the current title can be misleading, as it is not reflecting the theoretical nature of the paper. We will change the title as suggested.
>
> Best regards.

---

### Author Response · Authors · 2018-11-13
**On the practical relevance of our approach**

First of all, we would like to thank the three reviewers for their careful assessment of our paper. We are currently working on a new version of our article which includes several improvements addressing the reviewers' concerns. One issue that was raised by the three reviewers is related to the practicality of our approach and the interpretation of the overall complexity bound. We provide a justification and clarification here.

We would like to stress that one of the key points is that the LP size depends *linearly* on the sample size. For a given architecture the bounds are *polynomial* in 1/epsilon. Indeed, it is the linear dependence on the sample size in such a generic setting that is surprising as this provides strong justification why training scales nicely with the size of the data, as observed in practice. We expect that this insight will have applications beyond our LP approach. In addition, to the best of our knowledge, before our paper the best training complexity bounds become polynomial in the sample size only *after* fixing the architecture parameters (input dimension, depth, width, etc.), whereas ours is polynomial in the sample size irrespective of the architecture; we should have been more clear in the presentation of our results.

The reviewers are correct in that the LPs we are constructing are large and likely to be unsolvable by writing down our formulation directly and relying on off-the-shelf solvers. This fact would make our contribution seem, at this stage, impractical. However, the history of LPs shows that in many interesting cases, very large LPs can be solved to proved optimality or near-optimality while only generating a vanishingly small part of the LP. We might say that such algorithmic approaches follow an incremental strategy. The classical example is given by Edmonds' weighted matching algorithm. From a purely practical perspective, we have the LPs arising in the airline or mining industry, among others,  where (again) the LP is never actually written down. In all these cases one has an LP with a rich underlying structure, and the algorithms exploit that structure to solve the problem. In these examples and others, the theoretical understanding of the LP structure is used to drive the development of incremental solution strategies. In this paper, we have focused on laying a theoretical foundation.

To further expand on this point, note that the development of practical counterparts to theoretical algorithms can be a challenging task, and much effort is needed. In the case of Mixed-Integer Programming, for example, exact algorithms heavily rely on heuristics that do not have any guarantee, but that work well in practice and allow the algorithms to run faster without compromising overall exactness. Our hope is that our novel "polyhedral" interpretation of training can work together with state-of-art and practically efficient training algorithms (like stochastic gradient descent) in order to produce more complete training algorithms (perhaps with near optimality guarantees).  An idea that comes to mind would be a partially enumerative version of SGD based on our LP/IP.

Besides laying the foundations to what we think will become practical, we strongly believe that a theoretical understanding of the training problems is an important and necessary contribution to the ML community. For example, the way our approach works effectively implicitly "decomposes" the problem for each data point, and the LP merges them back together without losing any information nor optimality guarantee, even in this non-convex setting. This bears close resemblance to SGD where single data points (or batches of those) are used in a given step, and as such our results might provide a new perspective, through low treewidth, on *why* the current practical approaches work so well, as our LPs are working similarly.

Lastly, regarding the choice of outlet, we strongly consider ICLR to be a great fit for our paper as our work can be viewed as an improvement of the work in ICLR by Arora et al. (2018) in the bounded case, which posed several questions open that we answer here. Fundamental understanding of the training problem has recently gained increasing traction within the machine learning community. See, for example, the recent paper by Manurangsi and Reichman (arXiv:1810.04207; posted after ours). This paper analyzes the training problem in the special case of ReLUs, obtaining an algorithm with a worse dependency on epsilon than ours.

Best regards.

---

> ### Comment · Area_Chair1 · 2018-11-29
> **question**
>
> I'm confused by this sentence:
>
> " In addition, to the best of our knowledge, before our paper the best training complexity bounds become polynomial in the sample size only *after* fixing the architecture parameters (input dimension, depth, width, etc.), whereas ours is polynomial in the sample size irrespective of the architecture; we should have been more clear in the presentation of our results."
>
> A trivial brute-force search would give runtime exp(#parameters)*n. Would that be counted as "polynomial in the sample size irrespective of the architecture"? Maybe I missed something obvious?

---

> > ### Author Response · Authors · 2018-11-29
> > **clarification**
> >
> > The AC is right in pointing out that discretization already gives the desired dependence. Note that algorithms in previous papers achieve *worse* results than the brute-force method. Partly this is due to the fact that some of these algorithms do not assume boundedness of the parameters, but this might be a simple technical artifact. In [1], for example, the authors discuss cases when boundedness can be imposed without losing any classification performance. In the bounded case, the "brute-force" approach induced by our discretization method certainly achieves the desired dependence on the sample data; we agree it is somewhat surprising that this has not been observed beforehand (to the best of our knowledge). Under the discretization method one has to work out some error bound dependencies (which we do) and while this might be a nice insight in and of itself this is but a corollary of our main contribution.
> >
> > In fact, the key contribution is that our LP is *uniform* or *data-independent*; this is the same thing just different names from different communities. What this means is that the LP encodes all possible training problems (up to the approximation guarantee), while still keeping a linear dependence on the sample size. This is quite surprising and not something that is easily achievable using a brute-force approach. Why is this important? A helpful analogy is to compare to what one would require from an algorithm: it should work on a family of data sets and not just on a specific dataset. The uniformity requirement is the analog of this requirement for linear programs and as such natural and *necessary*, so that it does not just become brute-force enumeration for a *given* data set or trivial LPs.
> >
> > [1] Liao, Qianli, et al. "A surprising linear relationship predicts test performance in deep networks." arXiv preprint arXiv:1807.09659 (2018).

---

### Meta-Review · Area_Chair1 · 2018-12-17

**Confidence:** 5
**Recommendation:** Reject

**Metareview:**

The strength of the paper is that it designs an LP-based algorithm for training neural networks with runtime exponential in the number of parameters and linear in the size of the datasets and the algorithm works for worst-case datasets. As reviewer 2 and reviewer 3 pointed out, the cons include a) it's not clear why the algorithm provides any theoretical insights on how to design in the future polynomial time algorithm --- it seems that the algorithms are inherently exponential time and b) it's not clear whether the algorithm is practically at all relevant. The AC also noted that brute-force search algorithm is also exponential in # parameters and linear in size of datasets, and the authors agreed with it. This leaves the main contribution of the paper be that it works for the worst datasets. However, theoretically speaking, it's not clear this should be counted as a feature for algorithm design because we cannot go beyond the intractability without making assumptions on the data and in the AC's opinion, the big open question is how to make additional assumptions on the data (instead of removing them.) In summary, the drawback b) makes this a purely theoretical paper and the theoretical significance of the paper is unclear due to a). Therefore based on a), the AC decided to recommend reject, although the AC suggested the authors to re-submit to other top theory or ML theory conferences which may better evaluate the theoretical significance of the paper.